# Quantum MASALA: Quantum MAterialS Ab initio eLectronic-structure pAckage

Shri Hari Soundararaj, Agrim Sharma and Manish Jain⋆

Centre for Condensed Matter Theory, Department of Physics,
Indian Institute of Science, Bangalore 560012, India
⋆mjain@iisc.ac.in

August 15, 2023

## Abstract

We present `QuantumMASALA`, a compact package that implements different electronic-structure methods in Python. Within just 8000 lines of pure Python code, we have implemented Density Functional Theory (DFT), Time-dependent Density Functional Theory (TD-DFT) and the GW Method. The program can run across multiple process cores and in Graphical Processing Units (GPU) with the help of easily-accessible Python libraries. With `QuantumESPRESSO` and `BerkeleyGW` I/O interfaces implemented, it can also be used as a substitute for small scale calculations, making it a perfect learning tool for *ab intio* methods. The package is aimed to provide a framework with its modular and simple code design to rapidly build and test new methods for first-principles calculation.

# 1 Introduction

Over the last few decades, *ab initio* electronic structure calculations have become a widespread methodology for studying various properties of molecules and materials. The exponential growth of computation power along with its accessibility has made first-principle calculations ubiquitous in condensed matter studies. Due to its enormous success, density functional theory (DFT) [1,2] has become the standard technique for computational analysis of both molecular and periodic systems. The GW Method [3], although computationally more expensive, has been proven to describe excitations in systems with high accuracy [4].

The development of robust, efficient and massively-parallel computer codes that implement such techniques has massively contributed to their widespread use. DFT Codes like `QuantumESPRESSO` [5,6], `ABINIT` [7,8], `VASP` [9], `PARSEC` [10] and `Octopus` [11, 12] have been in active development for decades during which they evolved into feature-rich packages built to be efficient and massively scalable. Similarly, several packages such as `BerkeleyGW` [4] and `YAMBO` [13] have a massively parallel implementation of the GW method which has been demonstrated to solve systems containing thousands of atoms.

The recent developments in the field of data science have motivated the development of high-throughput screening of materials in order to accelerate the discovery of novel materials and structures. The application of machine learning to electronic structure calculations is also an active field of research. With the rising interests in studying larger and more complex materials in search of novel properties, there is a need to formulate new methods/algorithms

that improve the speed and efficiency of *ab initio* calculations, which is usually the performance bottleneck of such systems.

While being efficient and scalable, the aforementioned packages are not best suited for quickly 'prototyping' new ideas and algorithms. During the long development period of these packages, they have evolved into monoliths containing hundreds of lines of code, resulting in steep learning curve for newcomers to the package. The general lack of flexibility/customizability in the low-level routines leads to much slower rate of development.

To address the need for a compact framework, we have designed QuantumMASALA, a Python [14] package aimed to be a suite of electronic structure methods with a simple code design[1]. With less than 8000 lines of pure Python code, QuantumMASALA implements the pseudopotential based DFT [15], time-dependent DFT [11, 12] and GW method [16] for periodic solids. QuantumMASALA is designed with minimal required dependencies and supports multiple optional dependencies that enable MPI-parallelism and GPU-Acceleration, maximizing performance in any device. Written in one of the most popular programming language that is known for its readable syntax, the package aims to provide a sandbox for implementing and testing new ideas quickly.

This article gives a complete description of the QuantumMASALA package. We begin with a brief overview of the theory behind the implemented electronic structure methods. We describe the codebase, starting with a short discussion about runtime performance in Python. We discuss the general layout of the code which includes a brief description of the package's core modules and structures. The algorithms that are implemented in major calculation routines are described in detail. Finally, we demonstrate the capabilities of the package and provide an analysis of its performance and accuracy in comparison with popular packages. We conclude with a brief summary of the package's features and future development plans of the code.

## 2 Theoretical Background

### 2.1 Density functional theory and the Kohn-Sham equations

Density functional theory relates the ground-state electron density of a given system directly to the total energy. The Kohn-Sham formulation [2] of DFT maps the density of the many-body problem to an independent-particle system that is described by the Kohn-Sham (KS) Hamiltonian (in atomic units):

$$\hat{H}_{\text{KS}} |\psi_i\rangle = \left[ -\frac{1}{2}\nabla_i^2 + V_{\text{aux}}(\mathbf{r}; n(\mathbf{r})) \right] |\psi_i\rangle \tag{1}$$

$$n(\mathbf{r}) = \sum_i f_i |\psi_i(\mathbf{r})|^2 \tag{2}$$

where $f_i$ denotes the occupation number of the state $|\psi_i\rangle$.

The first term of $\hat{H}_{\text{KS}}$ is the single-particle kinetic energy operator. The local potential $V_{\text{aux}}(\mathbf{r}, n(\mathbf{r}))$ depends on the ground-state density $n(\mathbf{r})$ and is given by the following:

$$V_{\text{aux}}(\mathbf{r}; n(\mathbf{r})) = V_{\text{ext}}(\mathbf{r}) + V_{\text{Hartree}}(\mathbf{r}; n(\mathbf{r})) + V_{\text{xc}}(\mathbf{r}; n(\mathbf{r})) \tag{3}$$

where $V_{ext}$ is the external potential, which in the case of the electronic structure problems is the Coulomb potential generated by the nuclei. The Hartree potential is the classical potential on an electron due to charge density and the exchange-correlation (XC) potentials represents the remaining many-body interaction. Note that $V_{\text{xc}}$ can be spin-dependent based on the choice of the XC functional.

---

[1]QuantumMASALA GitHub repository: https://github.com/qtm-iisc/QuantumMASALA

The density dependence of the potential in $\hat{H}_{KS}$ requires the ground-state solution to be self-consistent with the potential in the Hamiltonian. Therefore, solving the KS system requires an self-consistent-field (SCF) iteration, which involves the following sequence of quantities to be evaluated:

$$n_1 \rightarrow V_1 \rightarrow \{|\psi_i\rangle\}_1 \rightarrow n_{\text{out},1} \xrightarrow[\text{mix}]{n_{\text{out},1} \neq n_1} n_2 \tag{4}$$

$$n_2 \rightarrow V_2 \rightarrow \{|\psi_i\rangle\}_2 \rightarrow n_{\text{out},2} \xrightarrow[\text{mix}]{n_{\text{out},2} \neq n_2} n_3 \tag{5}$$

$$\vdots \tag{6}$$

$$n_N \rightarrow V_N \rightarrow \{|\psi_i\rangle\}_N \rightarrow n_{\text{out},N} \xrightarrow{n_{\text{out},N} = n_N} n_N \equiv n_{\text{KS}} \tag{7}$$

The number of steps required for achieving self-consistency can be reduced using fixed-point iteration algorithms, which involves a mixing of densities of input, output, and in some methods, values from previous iterations too.

## 2.2 Wavefunctions in Periodic Solids and the Plane-Wave Basis

In periodic solids, according to Bloch's Theorem, the wavefunctions of the KS Hamiltonian can be written as:

$$|\psi_i\rangle \rightarrow |\psi^i_{\mathbf{k}}\rangle \, ; \, \langle\mathbf{r}|\psi^i_{\mathbf{k}}\rangle = \psi^i_{\mathbf{k}}(\mathbf{r}) := e^{i\mathbf{k}\cdot\mathbf{r}} u^i_{\mathbf{k}}(\mathbf{r}) \tag{8}$$

where $u^i_{\mathbf{k}}(\mathbf{r})$ has the periodicity of the crystal. The periodic function $u^i_{\mathbf{k}}(\mathbf{r})$ can be expressed as a Fourier series, using the plane wave expansion

$$\langle\mathbf{r}|\psi^i_{\mathbf{k}}\rangle = e^{i\mathbf{k}\cdot\mathbf{r}} \cdot \sum_{\mathbf{G}} u^i_{\mathbf{k}}(\mathbf{G}) e^{i\mathbf{G}\cdot\mathbf{r}} \tag{9}$$

$$= \sum_{\mathbf{G}} u^i_{\mathbf{k}}(\mathbf{G}) \langle\mathbf{r}|\mathbf{G}+\mathbf{k}\rangle \, ; \, \langle\mathbf{r}|\mathbf{G}+\mathbf{k}\rangle := e^{i(\mathbf{G}+\mathbf{k})\cdot\mathbf{r}} \tag{10}$$

where $\mathbf{G}$ are the lattice points of the reciprocal lattice of the crystal.

The KS Hamiltonian in the plane wave(PW) basis takes the following form:

$$\langle\mathbf{G}_m+\mathbf{k}|\hat{H}_{KS}|\mathbf{G}_n+\mathbf{k}\rangle = \langle\mathbf{G}_{m+\mathbf{k}}|\left[-\frac{1}{2}\nabla^2 + V_{KS}(\mathbf{r})\right]|\mathbf{G}_n+\mathbf{k}\rangle \tag{11}$$

$$H_{m,n}(\mathbf{k}) = \frac{|\mathbf{k}+\mathbf{G}_m|^2}{2}\delta_{m,n} + V_{KS}(\mathbf{G}_m-\mathbf{G}_n) \tag{12}$$

For solving the system numerically, the PW Basis is truncated. The wave functions are expanded including only those plane waves whose energy is less than a energy cutoff, $E_{\text{cut}}$ *i.e.* ($|\mathbf{k}+\mathbf{G}_m|^2/2 \le E_{\text{cut}}$). This implies that the PW expansion of $V_{KS}$ must contain $G$ within the energy cutoff $4E_{\text{cut}}$ as $|\mathbf{G}_m-\mathbf{G}_n| \le |\mathbf{G}_m| + |\mathbf{G}_n| \le 2\sqrt{2E_{\text{cut}}}$

## 2.3 Pseudopotentials:

Most chemical behaviour of atoms originate from the behaviour of the valence electrons which is influenced by their surrounding chemical environment. The core electrons, which remain unaffected under most situations, can be replaced with an effective 'pseudo-potential' on the valence electrons that can accurately reproduce their states across a wide range of conditions. As pseudopotentials aim to replace the potential generated by the nucleus and core electrons,

they must be spherically symmetric where each $(l, m)$ is treated individually. The 'semi-local' form of a pseudopotential of an atom centred at origin is given below:

$$\hat{V}_{ps} = \sum_l \sum_{m=-l}^l |l, m\rangle V^l(r) \langle l, m| \tag{13}$$

This form can be simplified to a sum of projection operators, as given by Kleinman and Bylander [17]:

$$\hat{V}_{ps} = V_{loc}(\mathbf{r}) + \sum_{(l,m)} \frac{\left|\Delta V_{ps}^l \varphi_{lm}\right\rangle\!\!\left\langle\Delta V_{ps}^l \varphi_{lm}\right|}{\langle\varphi_{lm}|\Delta V_{ps}^l|\varphi_{lm}\rangle}; \ \Delta V_{ps}^l = V^l - V_{loc} \tag{14}$$

## 2.4 Time-dependent Density Functional Theory:

The Time-Dependent Density Functional Theory [18] extends DFT from describing ground-state properties to evaluating the real-time dynamics, enabling it to compute excited state properties and response to time-dependent perturbations. The time-dependent Kohn-Sham (TDKS) equations take a form that is analogous to the time-dependent Schrodinger equation:

$$i\frac{\partial}{\partial t}|\psi_i\rangle = \hat{H}_{KS}(t)|\psi_i\rangle \tag{15}$$

$$= \left[-\frac{1}{2}\nabla^2 + V_{KS}(\mathbf{r}, t; n(\mathbf{r}, t))\right]|\psi_i\rangle \tag{16}$$

$$n(\mathbf{r}, t) = \sum_i f_i|\psi_i(\mathbf{r}, t)|^2 \tag{17}$$

The potential term $V_{KS}(\mathbf{r}, t)$ here is analogous to the time-independent equations, containing a time-varying external potential $V_{ext}(\mathbf{r}, t)$. For the XC Potentials in TDDFT, we limit ourselves to adiabatic approximations where $V_{xc}(\mathbf{r}, t; n(\mathbf{r}, t)) := V_{xc}(\mathbf{r}; n(\mathbf{r}, t))$

The formal solution of the TDKS Equations is given by:

$$|\psi_i(t)\rangle = \mathcal{T}\exp\left\{-i\int_0^t d\tau\, \hat{H}_{KS}(\tau)\right\}|\psi_i(t=0)\rangle \tag{18}$$

$$= \hat{U}_{KS}(t, 0)|\psi_i(t=0)\rangle \tag{19}$$

where $\mathcal{T}$ denotes the time-ordering operator. The time-evolution operator $\hat{U}(t', t)$ can be split into smaller time-steps:

$$\hat{U}(t_N, t_0) = \hat{U}(t_N, t_{N-1}) \cdot \hat{U}(t_{N-1}, t_{N-2}) \cdots \hat{U}(t_2, t_1) \cdot \hat{U}(t_1, t_0); \ t_n = t_0 + n\Delta t \tag{20}$$

TDDFT can be used to compute the optical absorption spectra by propagating the TDKS equations. By perturbing the ground-state system with a delta electric field along, lets say x-axis, $v_{pert}(\mathbf{r}, t) = -k_0\delta(t) \cdot (\mathbf{r} \cdot \hat{x})$, the system evolves in time with a density response $\delta n(\mathbf{r}, t) = n(\mathbf{r}, t) - n_0(\mathbf{r})$. For small perturbation $k_0 << 1$, the response is expected to be linear and dipolar, allowing us to compute the dynamical polarizability from the dipole response

$$\alpha_{x\nu}(\omega) = -\frac{1}{k_0}\int d^3r\, \delta n(\mathbf{r}, \omega)r_\nu; \ \ \nu = x, y, z \tag{21}$$

The absorption cross-section is proportional to the imaginary part of $\alpha$ averaged over the three spatial directions.

$$\sigma(\omega) = \frac{4\pi\omega}{c} \cdot \Im\mathfrak{m}\frac{1}{3}\sum_\nu \alpha_{\nu\nu}(\omega) \tag{22}$$

## 2.5 GW Approximation

Kohn-Sham DFT yields the correct ground state electron density, but the corresponding non-interacting single-particle orbitals need not have any direct physical interpretation [3,19,20]. Electron addition and removal energies are quasiparticle energies and its calculation requires many-body effects to be taken into account [3,16,21]. This can be done using Hedin's GW formalism [3], where one directly calculates the quasiparticle energies as poles of the single particle Green's function. Within this formalism, the exact many-body self-energy is expressed as a perturbation series in terms of dynamically screened Coulomb interaction.

While in principle, to calculate the poles of the single-particle Greens function one needs to solve the Dyson equation, often, the quasiparticle energies are computed as a single shot correction to the DFT energy eigenvalues. Within this approach the Greens function ($G_0$) and screened Coulomb interaction($W_0$) are constructed from DFT orbitals and energy eigenvalues. For a large variety of materials, the first iteration is good enough for calculating the quasiparticle energy spectrum [22]. This method of doing a single GW iteration is known as the $G_0W_0$ method.

Written explicitly, we are interested in solving

$$\left[ -\frac{1}{2}\hat{\nabla}^2 + \hat{V}_{\text{ext}} + \hat{V}_{\text{Hartree}} + \hat{\Sigma}(E_{n\mathbf{k}}) \right] \psi_{n\mathbf{k}}^{\text{QP}} = E_{n\mathbf{k}}\, \psi_{n\mathbf{k}}^{\text{QP}} \tag{23}$$

where $\hat{\Sigma}$ is the $G_0W_0$ dynamical self energy operator which is non-local, and non-Hermitian in general [16]. As a result, the quasiparticle energy eigenvalues $E_{n\mathbf{k}}$ are complex - their imaginary parts being inversely related to quasiparticle lifetimes. While calculating the quasiparticle energy spectrum, $E_{n\mathbf{k}}^{\text{QP}} = \text{Re}\,(E_{n\mathbf{k}})$ for quasiparticles with long lifetimes, it is common practice to solve the real part of eqn.23, while evaluating the self energy operator $\hat{\Sigma}(E)$ at $E_{n\mathbf{k}}^{\text{QP}}$, instead of $E_{n\mathbf{k}}$. Further, in a typical calculation, the DFT eigenvectors are sufficiently close to the quasiparticle eigenvectors [4] for the following approximation to be used:

$$E_{n\mathbf{k}}^{\text{QP}} = E_{n\mathbf{k}}^{\text{DFT}} + \left\langle \psi_{n\mathbf{k}}^{\text{DFT}} \left| \left( \hat{\Sigma}\left(E_{n\mathbf{k}}^{\text{QP}}\right) - V_{\text{xc}} \right) \right| \psi_{n\mathbf{k}}^{\text{DFT}} \right\rangle \tag{24}$$

The $G_0W_0$ self-energy $\Sigma$ is defined as [3]:

$$\Sigma\left(\mathbf{r},\mathbf{r}';t,t'\right) = i\, G_0\left(\mathbf{r},\mathbf{r}';t,t'\right) W_0\left(\mathbf{r},\mathbf{r}';t+0^+,t'\right) \tag{25}$$

where $G_0(\mathbf{r},\mathbf{r}';t,t')$ is the single particle Green's function

$$G_0\left(\mathbf{r},\mathbf{r}';t,t'\right) = -i\,\langle N|\, T\left[ \hat{\psi}^\dagger\left(\mathbf{r}',t'\right) \hat{\psi}\left(\mathbf{r},t\right)\right]|N\rangle \tag{26}$$

and $W_0(\mathbf{r},\mathbf{r}';t,t')$ is screened Coulomb interaction

$$W_0\left(\mathbf{r},\mathbf{r}';t,t'\right) = \int \epsilon^{-1}\left(\mathbf{r},\mathbf{r}'';t,t'\right) v\left(\mathbf{r}'',\mathbf{r}'\right) d\mathbf{r}'' \tag{27}$$

In time translation invariant systems, the self energy effectively becomes a function of $t-t'$ instead of $t$ and $t'$. The same holds for $G_0$, $W_0$, and $\epsilon$.

In the rest of the section we will use Rydberg units. In the plane wave basis, the random phase approximation (RPA) dielectric function is:

$$\epsilon_{\mathbf{GG}'}(\mathbf{q};\omega) = \delta_{\mathbf{GG}'} - v(\mathbf{q}+\mathbf{G})\, P_{\mathbf{GG}'}(\mathbf{q};\omega) \tag{28}$$

where $\mathbf{G}$ and $\mathbf{G}'$ are reciprocal lattice vectors, $\mathbf{q}$ lies in the first Brillouin zone, $v(\mathbf{q}+\mathbf{G})$ is bare Coulomb potential:

$$v(\mathbf{q}+\mathbf{G}) = \frac{8\pi}{|\mathbf{q}+\mathbf{G}|^2} \tag{29}$$

and $P$ is irreducible polarizability, for which the Adler-Wiser expression simplifies to [4]:

$$P_{\mathbf{GG'}}(\mathbf{q};\omega=0) = \sum_{n}^{\text{occ}}\sum_{n'}^{\text{emp}}\sum_{\mathbf{k}} \frac{\langle n'\mathbf{k}| e^{-i(\mathbf{q}+\mathbf{G})\cdot\mathbf{r}} |n\mathbf{k}+\mathbf{q}\rangle \langle n\mathbf{k}+\mathbf{q}| e^{i(\mathbf{q}+\mathbf{G'})\cdot\mathbf{r}} |n'\mathbf{k}\rangle}{E_{n\mathbf{k}+\mathbf{q}} - E_{n'\mathbf{k}}}. \tag{30}$$

In the GW formalism, the real part of self-energy as defined above naturally splits into two parts: the product of the real parts of $G_0$ and $W_0$ gives the Screened exchange (SX) term and the product of their imaginary parts gives the Coulomb-hole (CH) term. [23]

$$\text{Re}(\Sigma) = \Sigma_{\text{SX}} + \Sigma_{\text{CH}} \tag{31}$$

Under the static COHSEX approximation proposed by Hedin [3], the self-energy matrices can be expressed as:

$$\langle n\mathbf{k}| \Sigma_{\text{SX}}(E=0) |n'\mathbf{k}\rangle = -\sum_{n''}^{\text{occ}}\sum_{\mathbf{qGG'}} \langle n\mathbf{k}| e^{i(\mathbf{q}+\mathbf{G})\cdot\mathbf{r}} |n''\mathbf{k}-\mathbf{q}\rangle\langle n''\mathbf{k}-\mathbf{q}| e^{-i(\mathbf{q}+\mathbf{G'})\cdot\mathbf{r'}} |n'\mathbf{k}\rangle$$
$$\times \epsilon_{\mathbf{GG'}}^{-1}(\mathbf{q};0)v(\mathbf{q}+\mathbf{G'}) \tag{32}$$

$$\langle n\mathbf{k}| \Sigma_{\text{CH}}(E=0) |n'\mathbf{k}\rangle = \frac{1}{2}\sum_{n''}\sum_{\mathbf{qGG'}} \langle n\mathbf{k}| e^{i(\mathbf{q}+\mathbf{G})\cdot\mathbf{r}} |n''\mathbf{k}-\mathbf{q}\rangle\langle n''\mathbf{k}-\mathbf{q}| e^{-i(\mathbf{q}+\mathbf{G'})\cdot\mathbf{r'}} |n'\mathbf{k}\rangle$$
$$\times \left[\epsilon_{\mathbf{GG'}}^{-1}(\mathbf{q};0) - \delta_{\mathbf{GG'}}\right]v(\mathbf{q}+\mathbf{G'}) \tag{33}$$
$$= \frac{1}{2}\sum_{\mathbf{qGG'}} \langle n\mathbf{k}| e^{i(\mathbf{G}-\mathbf{G'})\cdot\mathbf{r}} |n'\mathbf{k}\rangle\left[\epsilon_{\mathbf{GG'}}^{-1}(\mathbf{q};0) - \delta_{\mathbf{GG'}}\right]v(\mathbf{q}+\mathbf{G'}) \tag{34}$$

We note that the completeness relation of mutually orthogonal orbitals has been used in (eq.34), to skip the expensive summation over all bands.

Also note that conventionally, the self energy is expanded as the sum of a Hartree-Fock exchange term and a (Coulomb) correlation term:

$$\Sigma = \Sigma_{\text{X}} + \Sigma_{\text{Corr}} \tag{35}$$

The Hartree-Fock exchange matrix elements can be calculated as follows:

$$\langle n\mathbf{k}| \Sigma_{\text{X}} |n'\mathbf{k}\rangle = -\sum_{n''}^{\text{occ}}\sum_{\mathbf{qGG'}} \langle n\mathbf{k}| e^{i(\mathbf{q}+\mathbf{G})\cdot\mathbf{r}} |n''\mathbf{k}-\mathbf{q}\rangle\langle n''\mathbf{k}-\mathbf{q}| e^{-i(\mathbf{q}+\mathbf{G'})\cdot\mathbf{r'}} |n'\mathbf{k}\rangle$$
$$\times \delta_{\mathbf{GG'}}v(\mathbf{q}+\mathbf{G'}) \tag{36}$$

The correlation term captures the effect of screening due to other electrons. We can identify the real part of the correlation term to be $\text{Re}(\Sigma_{\text{Corr}}) = (\Sigma_{\text{SX}} - \Sigma_{\text{X}}) + \Sigma_{\text{CH}}$.

Beyond Static-COHSEX, dynamical self-energy needs to be solved self-consistently, as shown below.

$$E_{n\mathbf{k}}^{\text{QP}} = \text{Re}(E_{n\mathbf{k}}) \tag{37}$$
$$= \text{Re}\left(\Sigma_{n\mathbf{k}}\left(E_{n\mathbf{k}}^{\text{QP}}\right)\right) + E_{n\mathbf{k}}^{\text{DFT}} - V_{\text{xc},n\mathbf{k}}^{\text{DFT}} \tag{38}$$

The Hybertsen-Louie plasmon pole model [16] assumes that contribution from a single plasmon dominates the pole structure of the inverse dielectric function. Using Kramers-Kronig relations and a generalized f-sum rule [16], one can get rid of all free parameters in the single

plasmon dielectric function [23], and the static dielectric function value can be used to generate values at all frequencies. With this approximation, we obtain the following self-energy matrix elements:

$$\langle n\mathbf{k}|\,\Sigma_{\mathrm{SX}}(E)\,\big|n'\mathbf{k}\rangle = -\sum_{n''}^{\mathrm{occ}}\sum_{\mathbf{qGG'}}\langle n\mathbf{k}|\,e^{i(\mathbf{q+G})\cdot\mathbf{r}}\,\big|n''\mathbf{k-q}\rangle\langle n''\mathbf{k-q}\big|\,e^{-i(\mathbf{q+G'})\cdot\mathbf{r'}}\,\big|n'\mathbf{k}\rangle \quad (39)$$

$$\times\left[\delta_{\mathbf{GG'}} + \frac{\Omega_{\mathbf{GG'}}^2(\mathbf{q})\,(1-i\tan\phi_{\mathbf{GG'}}(\mathbf{q}))}{\left(E-E_{n''\mathbf{k-q}}\right)^2 - \tilde{\omega}_{\mathbf{GG'}}^2(\mathbf{q})}\right] v(\mathbf{q+G'})$$

$$\langle n\mathbf{k}|\,\Sigma_{\mathrm{CH}}(E)\,\big|n'\mathbf{k}\rangle = \frac{1}{2}\sum_{n''}\sum_{\mathbf{qGG'}}\langle n\mathbf{k}|\,e^{i(\mathbf{q+G})\cdot\mathbf{r}}\,\big|n''\mathbf{k-q}\rangle\langle n''\mathbf{k-q}\big|\,e^{-i(\mathbf{q+G'})\cdot\mathbf{r'}}\,\big|n'\mathbf{k}\rangle \quad (40)$$

$$\times\frac{\Omega_{\mathbf{GG'}}^2(\mathbf{q})\,(1-i\tan\phi_{\mathbf{GG'}}(\mathbf{q}))}{\tilde{\omega}_{\mathbf{GG'}}(\mathbf{q})\big(E-E_{n''\mathbf{k-q}}-\tilde{\omega}_{\mathbf{GG'}}(\mathbf{q})\big)}\,v(\mathbf{q+G'})$$

where the plasmon mode parameters are defined as follows:
The effective bare plasma frequency:

$$\Omega_{\mathbf{GG'}}^2(\mathbf{q}) = \omega_{\mathrm{p}}^2\,\frac{(\mathbf{q+G})\cdot(\mathbf{q+G'})}{|\mathbf{q+G}|^2}\,\frac{\rho(\mathbf{G-G'})}{\rho(\mathbf{0})} \quad (41)$$

the GPP mode frequency:

$$\tilde{\omega}_{\mathbf{GG'}}^2(\mathbf{q}) = \frac{|\lambda_{\mathbf{GG'}}(\mathbf{q})|}{\cos\phi_{\mathbf{GG'}}(\mathbf{q})} \quad (42)$$

and the renormalized bare plasmon frequency:

$$|\lambda_{\mathbf{GG'}}(\mathbf{q})|\,e^{i\phi_{\mathbf{GG'}}(\mathbf{q})} = \frac{\Omega_{\mathbf{GG'}}^2(\mathbf{q})}{\delta_{\mathbf{GG'}}-\epsilon_{\mathbf{GG'}}^{-1}(\mathbf{q};0)} \quad (43)$$

where, $\rho$ denotes the electron charge density and $\omega_{\mathrm{p}}^2 = 4\pi\rho(\mathbf{0})e^2/m$ denotes the classical plasma frequency.

Divergent cases arising from the above equations are handled as per the prescription given in BerkeleyGW [4]. Due to the $n''$-dependence of $\epsilon_{\mathrm{HLPP}}$, it is not possible to get rid of the $n''$ summation over all orbitals in $\Sigma_{\mathrm{CH}}$ calculation. However, several methods have been devised to reduce the number of empty orbitals required to make this sum converge. Static remainder approach discussed in [24] is one such scheme.

$$\langle n\mathbf{k}|\,\Sigma_{\mathrm{CH}}^{\infty}(E)\,\big|n'\mathbf{k}\rangle = \langle n\mathbf{k}|\,\Sigma_{\mathrm{CH}}^{N}(E)\,\big|n'\mathbf{k}\rangle \quad (44)$$

$$+ \frac{1}{2}\bigg(\langle n\mathbf{k}|\,\Sigma_{\mathrm{CH}}^{\infty}(E=0)\,\big|n'\mathbf{k}\rangle - \langle n\mathbf{k}|\,\Sigma_{\mathrm{CH}}^{N}(E=0)\,\big|n'\mathbf{k}\rangle\bigg)$$

where $N$ or $\infty$ denote the number of empty orbitals included in $n''$ summation and '$E=0$' denotes static-COHSEX self-energies.

Dynamic self-energy calculated with plasmon-pole method depend on energy parameter $E$. In order to avoid a series of self-consistent iterations to compute $E_{n\mathbf{k}}^{\mathrm{QP}}$, we compute $\Sigma_{\mathrm{HLPP}}$ for two different sets of energies and then find $E_{n\mathbf{k}}^{\mathrm{QP}}$ using the secant method:

$$E_{n\mathbf{k}}^{\mathrm{QP}} = E_{n\mathbf{k}}^0 + \frac{d\Sigma/dE}{1-d\Sigma/dE}\left(E_{n\mathbf{k}}^0 - E_{n\mathbf{k}}^{\mathrm{MF}}\right) \quad (45)$$

where $E_{n\mathbf{k}}^0 = \Sigma_{\mathrm{HLPP}}\big(E_{n\mathbf{k}}^{\mathrm{MF}}\big)$ and $d\Sigma/dE$ is calculated using a forward finite difference scheme.

# 3 Writing performant code in Python: Libraries used

Python, as a programming language, is considered to be slow in comparison to compiled languages like C, C++, FORTRAN, etc. The reference implementation, CPython [14], is not as fast as other interpreted languages like MATLAB, Julia, R, etc. due to a lack of optimizations in its bytecode interpreter. High-throughput in Python is usually achieved by libraries and packages that have their core implementations written in 'faster' languages, usually C. The data structures and routines implemented can be accessed from Python via defined interfaces. For a Python program to be 'efficient' in comparison to its analogues written in a compiled program, it is important to maximize the amount of processing that can be performed in each call to optimised routines implemented in these libraries.

There are alternative strategies for speeding up Python. Some of the relevant ones are listed below:

- PyPy [25] is an alternative implementation of Python that implements Just-In-Time(JIT) Compilation to speed up execution.

- Numba [26] is a third-party package that brings JIT Compilation to CPython.

- Cython [27] extends Python with C-like syntax and compilation support that results in performance similar to a C/C++ code.

We have refrained from using the aforementioned optimization tools in order to maximize compatibility and reduce dependencies, but we encourage developers to try them when extending our package for their needs wherever applicable.

`QuantumMASALA` uses NumPy [28] for performing all linear algebra operations. The library provides support for multi-dimensional array of homogeneous data along with an extensive set of routines implementing mathematical operations and linear algebra algorithms. It is easy-to-use and interoperable across most scientific computing pacakges in Python. The library calls optimized BLAS and LAPACK functions for linear algebra operations, allowing access to near C/FORTRAN speeds in Python.

Analogous to NumPy, CuPy [29] is a Python library that contains implementations of a subset of NumPy operations to run in an NVIDIA GPU. It provides a NumPy-like interface to Arrays in GPU Memory with nearly identical function calls to operate on them. This enables `QuantumMASALA` to run computationally intensive part of a calculation like FFT's, diagonalization, etc. in a GPU. As a part of its design, `QuantumMASALA` reuses a lot of the CPU code for its GPU implementation, which is possible due to NumPy's interoperability with CuPy.

`QuantumMASALA` can also use FFT Libraries like FFTW3 (via `pyFFTW` [30]) and MKL (via `mkl_fft` [31]) if already installed. If not installed, the program automatically falls back to the slower FFT routines implemented in SciPy [32].

# 4 Usage: Core Objects of `QuantumMASALA`

## 4.1 Installation

The code for `QuantumMASALA` is available as a git repository at https://github.com/qtm-iisc/QuantumMASALA. The code supports Python version 3.9 and beyond. The installation of `QuantumMASALA` is as simple as doing a `pip install` at the root directory of the package.

```
pip install .
```

The user may choose to install with optional libraries for added features. MPI parallelization is provided by the library mpi4py [33, 34], and GPU support is provided by cupy [29]. The Python package primme provides interface to PRIMME [35], a C library for iterative solution of eigenvalue problems. The following command shows how to use `pip install` to install QuantumMASALA with these optional libraries.

```
pip install .[mpi4py, cupy, primme]
```

## 4.2 Defining a Crystal in QuantumMASALA

In QuantumMASALA, the lattice of translations is represented by the `Lattice` class. This class contains the primitive translation vectors and provides methods for coordinate transformation. The `RealLattice` and the `ReciLattice` classes extend the `Lattice` class with lattice parameter 'a' which is alat for `RealLattice` and tpiba($2\pi/a$) for `ReciLattice`. The two subclasses also provide methods to instantiate from instances of their duals. Some usage examples are highlighted below:

```python
import numpy as np

from quantum_masala.lattice import RealLattice, ReciLattice
from quantum_masala.constants import ANGSTROM

# Defining a BCC Lattice
alat = 5.1 * ANGSTROM  # Lattice parameter 'a'
# Lattice vectors
latvec_alat = 0.5 * np.array([
    [ 1,  1,  1],
    [-1,  1,  1],
    [-1, -1,  1]
])

# Creating RealLattice instance representing FCC lattice
reallat = RealLattice.from_alat(alat, *latvec_alat)
# Creating ReciLattice insance from reallat
recilat = ReciLattice.from_reallat(reallat)

# Printing axes
print(reallat.alat, reallat.axes_alat)
## 9.637603235591428 ([0.5, 0.5, 0.5], [-0.5, 0.5, 0.5], [-0.5, -0.5, 0.5])

print(recilat.tpiba, recilat.axes_tpiba)
## 0.6519447993019614 ([1.0, 0.0, 1.0], [-1.0, 1.0, 0.0], [0.0, -1.0, 1.0])

# Input array of vectors in crystal coordinates
vec_cryst = np.array([
    [1, 0, 0],
    [0, 1, 0],
    [1, 1, 0],
    [0, 1, 1]
])

# Coordinate Transforms
vec_cart = reallat.cryst2cart(vec_cryst, axis=1)
print(vec_cart)
```

```
## [[ 4.81880162  4.81880162  4.81880162]
##  [-4.81880162  4.81880162  4.81880162]
##  [ 0.          9.63760324  9.63760324]
##  [-9.63760324  0.          9.63760324]]

vec_alat = reallat.cryst2alat(vec_cryst, axis=1)
print(vec_alat)
## [[ 0.5  0.5  0.5]
##  [-0.5  0.5  0.5]
##  [ 0.   1.   1. ]
##  [-1.   0.   1. ]]

# NOTE: As per convention, the first dimension of vector lists
# must correspond to the components, resulting in a shape of
# form (3, ...)
len_alat = reallat.norm(vec_cryst.T, coords='cryst')
print(len_alat / reallat.alat)
## [0.8660254  0.8660254  1.41421356 1.41421356]
```

A crystal can be fully specified by the lattice and the atoms in the unit cell. Similarly, the `Crystal` class in `QuantumMASALA` is defined by:

1. A `RealLattice` instance describing the lattice of the crystal

2. A sequence of `AtomBasis` instances describing the crystal's basis atoms

The `AtomBasis` class represent a collection of atoms in the unit cell belonging to one atomic species/type and describes the position of the atoms in the crystal's unit cell along with the species' name, mass, and optionally its pseudopotential. Note that for DFT calculations, the pseudopotential is a required argument. Currently `QuantumMASALA` supports the UPF v2 format only via the `UPFv2Data` container.

```python
# Generating Crystal instance representing Iron
from quantum_masala.lattice import RealLattice
from quantum_masala.crystal import BasisAtoms, Crystal
from quantum_masala.pseudo import UPFv2Data

#from quantum_masala.utils import print_crystal_info
from quantum_masala.constants import BOHR

# Defining the lattice
reallat = RealLattice.from_alat(alat=5.1070 * BOHR,
                                a1=[ 0.5,  0.5,  0.5],
                                a2=[-0.5,  0.5,  0.5],
                                a3=[-0.5, -0.5,  0.5])

# Reading pseudopotential data
fe_oncv = UPFv2Data.from_file('fe', 'Fe_ONCV_PBE-1.2.upf')
# Specifying the basis atoms
fe_atoms = BasisAtoms.from_cart(
    'fe', 55.487, fe_oncv, reallat,
    [0., 0., 0.]
)
# Constructing the Crystal instance
crystal = Crystal(reallat, [fe_atoms, ])
```

```
print_crystal_info(crystal)
```

**Output**

```
Lattice parameter 'alat' :    5.10700   a.u.
Unit cell volume          :   66.59898  (a.u.)^3
Number of atoms/cell      : 1
Number of atomic types    : 1
Number of electrons       : 16

Crystal Axes: coordinates in units of 'alat' (5.10700 a.u.)
    a(1) = (  0.50000,    0.50000,   0.50000)
    a(2) = ( -0.50000,    0.50000,   0.50000)
    a(3) = ( -0.50000,   -0.50000,   0.50000)

Reciprocal Axes: coordinates in units of 'tpiba' (1.23031 (a.u.)^-1)
    b(1) = (   1.00000,   0.00000,   1.00000)
    b(2) = (  -1.00000,   1.00000,   0.00000)
    b(3) = (   0.00000,  -1.00000,   1.00000)

Atom Species #1
    Label   : fe
    Mass    : 55.49
    Valence : 16.00
    Pseudpot: Fe_ONCV_PBE-1.2.upf
             MD5: 0a0a3b4237ca738a29386c4fd1f8ac5e
    Coordinates (in units of alat)
         1 - (  0.00000,   0.00000,   0.00000)
```

### 4.3 Bases in Plane-Wave Codes: Real-Space and G-Space

Contrary to the name, Plane-Wave Codes perform calculations in both frequency (Fourier) space and real space. Although the former gives a compact representation of Plane-Waves, the latter is more effective for evaluating certain operations. For instance, a local potential has a simple operator representation in real-space, where it forms a diagonal matrix. But, the same in Fourier basis would yield a dense matrix due to convolution.

$$\hat{V} := \int_\Omega \mathrm{d}\mathbf{r}\, V(\mathbf{r})\, |\mathbf{r}\rangle\langle\mathbf{r}|$$

$$\langle\mathbf{G}|\hat{V}|\mathbf{G}'\rangle = \int_\Omega \mathrm{d}\mathbf{r}\, V(\mathbf{r})\, \langle\mathbf{G}|\mathbf{r}\rangle\, \langle\mathbf{r}|\mathbf{G}'\rangle$$

$$= \int_\Omega \mathrm{d}\mathbf{r}\, V(\mathbf{r}) \exp\{-i\mathbf{G}\cdot\mathbf{r}\} \exp\{i\mathbf{G}'\cdot\mathbf{r}\}$$

$$= \sum_{\mathbf{G}''} \int_\Omega \mathrm{d}\mathbf{r}\, V(\mathbf{G}'') \exp\{i\mathbf{G}''\cdot\mathbf{r}\} \exp\{-i(\mathbf{G}-\mathbf{G}')\cdot\mathbf{r}\}$$

$$= \sum_{\mathbf{G}''} V(\mathbf{G}'')\delta_{\mathbf{G}-\mathbf{G}',\mathbf{G}''}$$

Thanks to Fast Fourier Transforms, it is easier and more efficient to implement the operator in the following way:

1. Inverse Fourier Transform the Plane-Wave wave functions

$$\psi_{i,\mathbf{k}}(\mathbf{G}) \xrightarrow{IFFT} \psi_{i,\mathbf{k}}(\mathbf{r}) \tag{46}$$

2. Apply the real-space potential by simple element-wise multiplication

$$\hat{V}\left|\psi_{i,\mathbf{k}}\right\rangle \equiv \int_{\Omega} d\mathbf{r}\,|\mathbf{r}\rangle\langle\mathbf{r}|\,V(\mathbf{r})\psi_{i,\mathbf{k}}(\mathbf{r}) \tag{47}$$

3. Forward Fourier Transform the result

$$\left[\hat{V}\psi_{i,\mathbf{k}}\right](\mathbf{r}) \xrightarrow{FFT} \left[\hat{V}\psi_{i,\mathbf{k}}\right](\mathbf{G}) \tag{48}$$

Similarly, certain quantities such as the kinetic energy operator, the Hartree potential, etc. are simpler in Fourier space than in real-space. So, they can be evaluated in Fourier space as is and can be transform to real-space if needed.

$$\left\langle\mathbf{q}\middle|\hat{K}\middle|\mathbf{q}'\right\rangle = \left\langle\mathbf{q}\middle|-\frac{1}{2}\nabla^2\middle|\mathbf{q}'\right\rangle \tag{49}$$

$$= \frac{1}{2}\delta_{\mathbf{q},\mathbf{q}'}|\mathbf{q}|^2 \tag{50}$$

$$\left\langle\mathbf{q}\middle|\hat{K}\psi\right\rangle = \frac{\hbar^2 q^2}{2m_e}\psi(\mathbf{q}) \tag{51}$$

### 4.3.1 Truncation of Fourier Space: G-Space

The Kinetic energy operator in Fourier basis is $\frac{q^2}{2}|\mathbf{q}\rangle\langle\mathbf{q}|$ and the plane waves $|\mathbf{q}\rangle$ form the operator's eigenbasis. With the eigenvalues given by $\frac{q^2}{2}$, this operator will dominate the Hamiltonian beyond a certain $q$, translating to lowest energy eigenkets having near negligible components of large $|\mathbf{q}\rangle$ plane-waves. Therefore, an energy cutoff $E_{\text{cut}}$ is chosen to truncate the Fourier space to within the sphere $q \leq \sqrt{2E_{\text{cut}}}$. This truncated Fourier space will be referred to as **G-space** from here on as it contains the lattice points of the reciprocal lattice which are commonly referred to as G-vectors. Although the basis is truncated, the actual evaluation of Fourier Transforms would still rely on FFT operations involving a fixed regular grid. But, the components outside of the cutoff will be discarded.

### 4.3.2 Implementation: GSpaceBase

Transformation of quantities such as wavefunctions, charges, potentials, etc. between the two basis is one of the most commonly used operations in Plane-Wave Codes. Therefore, an easy-to-use and robust mechanism to transform between the two basis is critical for QuantumMASALA. With multiple FFT libraries available to use in Python, there is also a need for the implementation to be modular. In QuantumMASALA, the GSpaceBase class fully encapsulates the previously discussed FFT operation within a truncated Fourier space. Instantiated using a list of G-vectors defining the G-Space, it implements the following methods:

- GSpaceBase.create_buffer() Creates arrays that are optimal for the FFT routines. Supports library-specific constraints

- GSpaceBase.check_buffer() Ensures that the arrays are of correct shape and type and is suitable for the selected FFT Libaray

- GSpaceBase.r2g() Forward Fourier Transform to truncated G-Space

- `GSpaceBase.g2r()` Inverse Fourier Transform from G-Space to Real-Space

The `GSpace` class wraps `GSpaceBase` with an initialization routine that generates the list of G-vectors from an input Kinetic Energy Cutoff.

```python
import numpy as np

from quantum_masala.lattice import ReciLattice
from quantum_masala.gspace.gspc import GSpace
from quantum_masala.constants import RYDBERG

ecutwfn = 40 * RYDBERG
ecutrho = 4 * ecutwfn

# Creating a GSpace instance with KE cutoff E_cut = 160 Ry
recilat = ReciLattice.from_reallat(reallat)
gspc = GSpace(recilat, ecutrho)
print(gspc.grid_shape, gspc.size_g, gspc.size_r)
## (18, 18, 18) 2243 5832

# Printing the G-vectors of gspc
with np.printoptions(
    edgeitems=5, threshold=100, formatter={
    'float': lambda num: f'{num:5.1f}',
    'int': lambda num: f'{float(num):5.1f}',
}):
    for i in range(3):
        print(f'G_{i+1}: ', gspc.g_cryst[i])
    print(f'G^2: ', gspc.g_norm2)
## G_1:  [  0.0   0.0   0.0   0.0   0.0 ...  -1.0  -1.0  -1.0  -1.0  -1.0]
## G_2:  [  0.0   0.0   0.0   0.0   0.0 ...  -1.0  -1.0  -1.0  -1.0  -1.0]
## G_3:  [  0.0   1.0   2.0   3.0   4.0 ...  -5.0  -4.0  -3.0  -2.0  -1.0]
## G^2:  [  0.0   3.0  12.1  27.2  48.4 ...  78.7  51.5  30.3  15.1   6.1]

# Creating a array representing a random value scalar field
sfield_r = gspc.create_buffer_r(1)
sfield_r[:] = np.random.rand(gspc.size_r) \
    + 1j * np.random.rand(gspc.size_r)

# Transforming to G-Space using gspc instance
sfield_g = gspc.r2g(sfield_r)

# The above is equivalent to 3D forward FFT followed by
# taking components corresponding to reciprocal
# lattice points that are within the energy cutoff
print(np.allclose(
    sfield_g, np.fft.fftn(
        sfield_r.reshape(gspc.grid_shape)
    ).take(gspc.idxgrid)
))
## True
```

### 4.3.3 Optimizing FFT operations for G-Space

Implementing Fourier Transforms to and from the G-Space using a single 3D FFT operation might not be optimal as some Fourier components can be ignored. By the definition of G-Space,

certain regions of the FFT grid are set to zero and this region is fixed for a given lattice and cutoff parameters. When computing the 3D FFT of let's say a $N \times N \times N$ array, we can evaluate them with $N^2$ 1D FFT's across each of the three axes. Knowing the truncated region, we can skip calling the actual 1D FFT routines on the 1D 'sticks' that contain no non-zero elements.

In theory, this reduces the total number of FFT operations as long as the G-vectors are compactly grouped within the FFT Grid, which is the case when truncating with an energy cutoff. But, in practice, the implementation requires data to be rearranged between each 1D FFT calls. Also, 3D FFT implementations involve batched FFT calls where the library operates on multiple 1D arrays simultaneously. These two factors lowers the method's overall efficiency and limits the algorithm effectiveness to only large grid sizes.

The runtime performance is sensitive to its implementation details and highly dependent on the structure of the G-Space. But for wave-functions where the number of G-vectors is much smaller than the total number of FFT Grid points, this method reduces the computation involved in `r2g()`/`g2r()` methods.

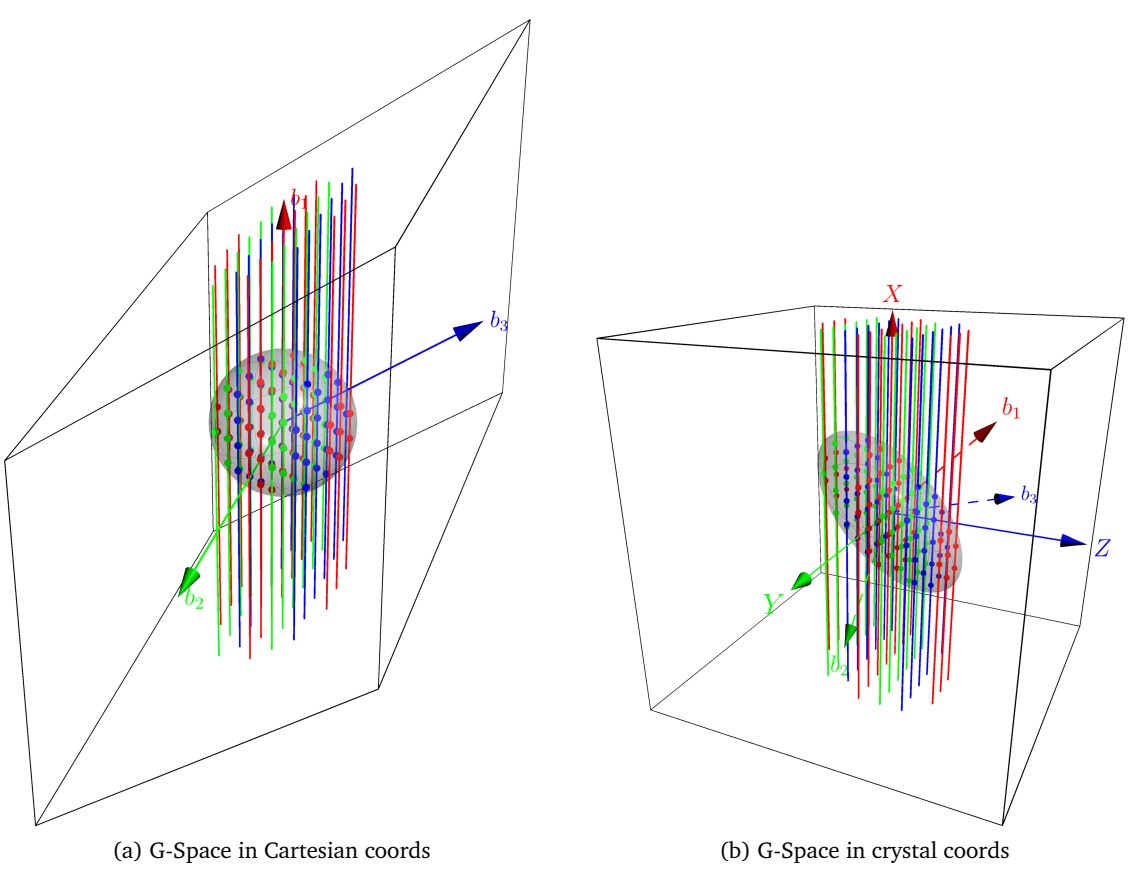

(a) G-Space in Cartesian coords          (b) G-Space in crystal coords

Figure 1: A Visual Representation of the G-Space of a BCC Iron lattice with lattice parameter 5.107 a.u. and Energy cutoff 25 Ry. FFT along X-Axis is performed only along the sticks that contain atleast one G-vector within cutoff energy

## 4.4 Data Containers: `Buffers`, `Fields` and `Wavefun`

For ease-of-use, we need to pair `GSpace` with a data container for representing and operating on with quantities like potentials, charge-density etc. that share the same periodicity as the crystal. A simple NumPy array would not be sufficient as it will not contain any data regarding the lattice and the G-Space, which is required for many methods such as computing the Hartree potential, evaluating the del operator on a scalar/vector field, constructing the kinetic energy

operator, etc. But, the container should mimic a lot of NumPy features like array slicing, indexing, broadcasting, list comprehension, etc. and binary operations like +, -, *, /, % etc.

Therefore, we have implemented the `Buffer` class that is a custom array-like container that enables the aforementioned functionalities. Initialized with a `GSpace` instance and NumPy array, it also provides FFT routines to convert between the two bases. The `Buffer` class is extended by three pairs of classes:

- `FieldR` and `FieldG`: Represents multidimensional fields

- `WavefunR` and `WavefunG`: Represents single particle bloch wavefunctions for a given k-point and stores its periodic part.

- `WavefunSpinR` and `WavefunSpinG`: similar to the Wavefun classes but incorporates spin, resulting in a doubling of basis sizes.

The Base `Buffer` class implements the following methods:

- `copy()`: creating copies of existing instances

- `empty(gspc, shape)` and `zeros(gspc, shape)`: Creating an empty/zero array with basis defined by `gspc` and input shape.

- `to_r()` / `to_g()`: conversion from G-Space to R-Space and vice-versa via respectively.

Internally, `Buffer` supports a select set of NumPy functions that operate on one element of an array at a time. Called as Universal Functions in NumPy, they provide a wide range of routines of which the `Buffer` class supports the following:

- Unary operations like computing trigonometric functions, negation, conjugation, logical/bit-wise inversion, etc.

- Binary operations such as +, -, *, /, % and other algebraic, logical and bit-wise operations.

- Reduction operations where one of the axes of the input array is reduced by repeatedly applying a binary operation to the values in array along the given axis.

- Matrix multiplication (only across the last two dimensions of array(s))

To improve performance, the `WavefunG` implements the `WavefunG.vdot(psi, phi)` method for evaluating the dot product $\langle \psi_i | \varphi_j \rangle$ that directly calls the ZGEMM BLAS method in `scipy.linalg.blas` module. This is to prevent calling `ndarray.conj()` method in NumPy that results in creating a new array and increasing memory usage.

```python
from quantum_masala.containers import FieldR, FieldG

def print_info(f):
    typ = type(f)
    basis = getattr(f, 'basis_type', 'N\A')
    data = getattr(f, 'data', None)
    shape = data.shape if data is not None else 'N/A'
    print(f"type: {str(type(f)):35}\n"
          f"basis: '{basis}', data.shape: {shape}\n")

# Creating a empty GField instance
a_g = FieldG.empty(gspc, (4, 10))
print_info(a_g)
```

```
## type: <class 'quantum_masala.containers.field.FieldG '>
## basis: 'g', data.shape: (4, 10, 2243)

# Converting to RField instance
a_r = a_g.to_r()
print_info(a_r)
## type: <class 'quantum_masala.containers.field.FieldR '>
## basis: 'r', data.shape: (4, 10, 5832)

# Indexing
arr_idx = a_r[2]
print_info(arr_idx)
## type: <class 'quantum_masala.containers.field.FieldR '>
## basis: 'r', data.shape: (10, 5832)

# Slicing
arr_sl = a_r[:, :5]
print_info(arr_sl)
## type: <class 'quantum_masala.containers.field.FieldR '>
## basis: 'r', data.shape: (4, 5, 5832)

# Unpacking
x, y = a_r[0, :2]
print_info(x)
print_info(y)
## type: <class 'quantum_masala.containers.field.FieldR '>
## basis: 'r', data.shape: (5832,)
## type: <class 'quantum_masala.containers.field.FieldR '>
## basis: 'r', data.shape: (5832,)

# Binary operations
z = x + y
print_info(z)
## type: <class 'quantum_masala.containers.field.FieldR '>
## basis: 'r', data.shape: (5832,)

# Binary operations with broadcasting
z = x + a_r[:2,:3]
print_info(z)
## type: <class 'quantum_masala.containers.field.FieldR '>
## basis: 'r', data.shape: (2, 3, 5832)

# Operations between mismatched basis; raises exception
z = x + x.to_g()
## TypeError: mismatch in 'basis_type ' between two 'Buffer ' instances.
```

## 4.5   GPU Acceleration in `QuantumMASALA`: Writing device-agnostic routines

Although GPUs has a much higher theoretical compute performance than CPU, porting CPU code to GPU is not easy, simply due to the difference in their architectures. For most applications, there is a need to write specialized code that is optimized for running in GPU, resulting in two separate code bases for running in each device. But, in Python, CuPy (GPU) provides an NumPy-like (CPU) interface to most linear algebra routines, allowing users to quickly port their NumPy codes for GPU Acceleration. This, paired with the Python array API standard [36], enables NumPy routines to work with GPU Arrays by implicitly calling the corresponding CuPy

routines. For instance, the keyword argument `like` in NumPy array creation method can be used to create arrays of the same type as the value passed to the argument, allowing us to write routines using NumPy functions that is also compatible with input CuPy arrays.

With this, `QuantumMASALA` aims to maximize the reuse of code for GPU routines and one of the key mechanisms implemented to achieve this goal is baked in the `GSpace` class; handling buffer creation on device and performing FFT operations on device. With a GPU implementation of this class `GSpaceCuPy`, one can reuse routines that contains CuPy-aware NumPy functions and perform the same operation(s) in the GPU by just passing the GPU instance. For instance, passing the `GSpaceCuPy` analog to `Buffer` constructors will create an instance with its data directly on GPU.

```python
import cupy as cp
from quantum_masala.gpu.gspace import GSpaceCuPy
from quantum_masala import qtmconfig

# This flag must be set to True to enable GPU usage in QuantumMASALA
qtmconfig.use_gpu = True

# Initializing GPU instance of 'gspc'
gspc_gpu = GSpaceCuPy(gspc)

field_gpu_r = FieldR.zeros(gspc_gpu, 1)
field_gpu_r.r[:] = cp.array(sfield_r)
print(type(field_gpu_r), type(field_gpu_r.r))
## <class 'quantum_masala.containers.field.FieldR'> <class 'cupy.ndarray'>

field_gpu_g = field_gpu_r.to_g()
np.allclose(field_gpu_g.data, sfield_g)
## array(True)

a_g = FieldG.empty(gspc_gpu, (4, 10))
print_info(a_g)
## type: <class 'quantum_masala.containers.field.FieldG'>
## basis: 'g', data.shape: (4, 10, 2243)

x, y = a_g[:, 0], a_g[:, 1]
z = x * y
print_info(z)
## type: <class 'quantum_masala.containers.field.FieldG'>
## basis: 'g', data.shape: (4, 2243)
```

## 4.6  Parallelization in `QuantumMASALA`

`QuantumMASALA` utilizes Message Passing Interface (MPI) routines provided by the `mpi4py` Python library to run across multiple processor cores. It can scale across hundreds of processor cores and contains multiple parallelization schemes for effectively distributing its calculation in parallel. Although `QuantumMASALA`'s implementations are written for parallel operations, it can still be run serially without installing `mpi4py`.

In order to achieve that, we have wrapped the core objects and the required operations in `mpi4py`, allowing the library to be an **optional** run-time dependency. We begin with a brief description about the wrapper and its usage.

### 4.6.1 `QTMComm`: **Wrapping MPI Communicators**

'Communicator' is the key object in MPI that connects groups of processes and provides an interface for communicating data across them. In `mpi4py`, this is encapsulated by `mpi4py.MPI.Comm` instances that are generated from a given group of processes. These instances provide methods for communicating data across members of the corresponding group. In MPI, the group containing all processes is called the 'WORLD' and its corresponding communicator is accessed by `mpi4py.MPI.COMM_WORLD` in `mpi4py`.

In order to write parallel code while maintaining the status of the library as an optional dependency, we have implemented a wrapper class named `QTMComm` that has a similar interface to `mpi4py.MPI.Comm`. It only imports `mpi4py` if available and if not available, it assumes serial operation and the wrapper will not utilize MPI routines at all while providing consistent behaviour. We have also extended the class to provide a few group operations such as `MPI_Group_split` and `MPI_Group_incl`.

We have also implemented context managers that serve the following purpose:

- It provides a visual cue that indicates parallelism i.e code within a `with` block show that the operation is across multiple processes.

- At the end of the `with` block, the Barrier operation `MPI_Comm_barrier` is called automatically, synchronizing all processes in the communicator

- The group operations are implemented with an additional boolean argument `sync_with_parent` that specifies if the barrier operation is also applied to the parent group when exiting the `with` block.

The last feature is especially handy in situations where you need only a subgroup of processes to execute a code-block while the remainder waits for their completion. Below, we will illustrate some use cases of `QTMComm`'s context manager.

### 4.6.2 **Example 1: Using** `QTMComm`**'s Context Manager**

```python
import time

from mpi4py.MPI import COMM_WORLD
from quantum_masala.mpi import QTMComm

# Creating 'QTMComm' instance from 'COMM_WORLD'
comm_world = QTMComm(COMM_WORLD)
wrld_size, wrld_rank = comm_world.size, comm_world.rank

# Function to add timestamp before printing to stdout
def print_msg(msg: str):
    curr_time = time.strftime("%H:%M:%S", time.localtime())
    print(f"{curr_time}: {msg}")

with comm_world as comm:
    size, rank = comm.size, comm.rank
    print_msg(f"Hello from process #{rank}/{size}")
    # Proc 1 going to sleep while the rest skip ahead
    if comm.rank == 1:
        print_msg(f"process #{rank} going to sleep for 3 seconds")
        time.sleep(3)
    print_msg(f"process #{rank} is at the end of 'with' code-block")
```

```
# When exiting, all procs in comm will be in sync.
# No need for 'comm.Barrier()' here.
# So all procs will print the message below at the exact time.
print_msg(f"process #{rank} has exited 'with' code-block")
```

**Output**

```
16:21:33: Hello from process #0/4
16:21:33: process #0 is at the end of 'with' code-block
16:21:33: Hello from process #1/4
16:21:33: process #1 going to sleep for 3 seconds
16:21:33: Hello from process #2/4
16:21:33: process #2 is at the end of 'with' code-block
16:21:33: Hello from process #3/4
16:21:33: process #3 is at the end of 'with' code-block
16:21:36: process #1 is at the end of 'with' code-block
16:21:36: process #0 has exited 'with' code-block
16:21:36: process #2 has exited 'with' code-block
16:21:36: process #1 has exited 'with' code-block
16:21:36: process #3 has exited 'with' code-block
```

### 4.6.3 Example 2: Dividing Communicator using `Split`

```
# Processes are divided into two groups
# rank     0  1  2  3
colors = [0, 1, 1, 0]  # Group number
keys   = [1, 1, 0, 0]  # Rank in group

wrld_size = comm_world.size
wrld_rank = comm_world.rank

c = colors[wrld_rank]
k = keys[wrld_rank]

with comm_world.Split(c, k) as comm:
    grp_id = comm.id_
    grp_size, grp_rank = comm.size, comm.rank
    print(f"process #{wrld_rank}/{wrld_size} is assigned to "
          f"subgroup #{grp_id} and its rank is "
          f"{grp_rank}/{grp_size}")
```

**Output**

```
process #0/4 is assigned to subgroup #0 and its rank is 1/2
process #1/4 is assigned to subgroup #1 and its rank is 1/2
process #2/4 is assigned to subgroup #1 and its rank is 0/2
process #3/4 is assigned to subgroup #0 and its rank is 0/2
```

### 4.6.4   Example 3: Running a code-block on a subset of processes using `Init`

```python
# Selecting the processess to include in subgroup
grp_iproc = [0, 3]

with comm_world.Incl(grp_iproc) as comm:
    if not comm.is_null:
        # The newly created group will be active
        grp_size, grp_rank = comm.size, comm.rank
        print_msg(f"process #{wrld_rank}/{wrld_size} is part of the "
                  f"subgroup and its rank is {grp_rank}/{grp_size}")
        print_msg(f"process #{wrld_rank} going to sleep for 3 seconds")
        time.sleep(3)
    else:
        # while the remaining processes wait at the end of the
        # 'with' codeblock
        print_msg(f"process #{wrld_rank}/{wrld_size} is not part of "
                  "the subgroup")
    print_msg(f"process #{wrld_rank} is at the end of 'with' code-block")

# When exiting, all procs in comm will be in sync.
# No need for 'comm.Barrier()' here.
# So all procs will print the message below at the exact time.
print_msg(f"process #{comm_world.rank} has exited 'with' code-block")
```

**Output**

```
14:42:30: process #1/4 is not part of the subgroup
14:42:30: process #2/4 is not part of the subgroup
14:42:30: process #1 is at the end of 'with' code-block
14:42:30: process #2 is at the end of 'with' code-block
14:42:30: process #3/4 is part of the subgroup and its rank is 1/2
14:42:30: process #0/4 is part of the subgroup and its rank is 0/2
14:42:30: process #3 going to sleep for 3 seconds
14:42:30: process #0 going to sleep for 3 seconds
14:42:33: process #3 is at the end of 'with' code-block
14:42:33: process #0 is at the end of 'with' code-block
14:42:33: process #0 has exited 'with' code-block
14:42:33: process #3 has exited 'with' code-block
14:42:33: process #2 has exited 'with' code-block
14:42:33: process #1 has exited 'with' code-block
```

### 4.6.5   Implemented Parallelization Modes

Using `QTMComm`, `QuantumMASALA` implements different levels of parallelization, which can be labelled as follows:

- Parallelism over **k-points**: Quantities that depend on only one or two k-points can be evaluated in parallel by distributing the k-points across subgroups called k-groups. Each k-group is assigned a subset of k-points/k-point pairs and are allowed to run independently till they are done with their assigned tasks. Communication across different k-groups is required only when the quantity is summed/reduced across k-points.

- Parallelism over **bands**: Subroutines that operate on a set of bands such as solving the Kohn-Sham Hamiltonian, computing the electron density from Bloch states, etc. usually involves operations that are applied on a batch of wave-functions which can be done in parallel like, for instance, evaluating $\hat{H}_{KS} |psi\rangle$. The k-group is further split into band-groups, which are assigned a subset of wave-functions to operate one. It must be noted that this parallelism is communication and usually memory intensive.

- Parallelism over **PW Basis**: This parallelism involves the distribution of the Plane-Wave basis/G-Space and its Real-Space dual. Implementation requires distributed FFT routines and frequent transfer of data between processes, making it the most communication-heavy scheme. But, unlike the band-distributed parallelization, the memory is distributed across processes, making it effective for large-size crystals and supercell calculations. Currently, this implementation is under testing.

## 5   Code layout

Apart from the core structures discussed in the previous section, the implementation of electronic-structure methods in `QuantumMASALA` is split into different sub-directories based on its scope, each forming its own submodules. In the following section, each module and its contents will be described in brief.

### 5.1   'pseudo' module:

The `quantum_masala.pseudo` module deals with constructing the (pseudo)-potential in the crystal from the given pseudopotential data. This potential represents the interaction between the valence electrons and the ions (nucleus + core electrons) in the crystal. `QuantumMASALA` implements the Norm-Conserving PseudoPotential (NCPP) and supports Optimized Norm-Conserving Vanderbilt (ONCV) Pseudpotentials [37] which are relatively simpler to implement compared to projector-augmented waves (PAWs) [38] and ultrasoft pseudopotentials [39], while yielding accurate results [40–42]

The pseudopotentials are evaluated in two parts:

- the local term generated using function `loc_generate_pot_rhocore` which also gives density of core electrons for non-local core correction (NLCC) in computing XC potentials.

- the non-local term generated using instances of class `NonlocGenerator`

This module also contains function
`loc_generate_rhoatomic` which is used by `quantum_masala.core.rho_atomic` to construct electron densities from superposition of atomic charges.

Currently, `QuantumMASALA` supports Unified Pseudopotential Format (UPF) v2 files as input, which is processed when specifying the basis atoms of crystal using `quantum_masala.core.BasisAtoms`. The SG15 [43] ONCV set of psuedopotential is recommended as it has been extensively tested with `QuantumMASALA` as part of the DFT benchmarks.

### 5.2   'dft' module:

The `dft` module of `QuantumMASALA` implements the solvers of the KS Hamiltonian and the SCF iteration routines.

For computing the ground-state of the KS system, only the lowest occupied states of the Hamiltonians are required, which constitutes a small fraction of the eigenvalues of the Hamiltonian Matrix (11). `QuantumMASALA` currently implements the Davidson Method with Overlap, which is based on the same from `QuantumESPRESSO`. A key feature of these methods is that they do not require the explicit construction of the matrix that is being diagonalized. The routines instead involve repeated application of the matrix on 'guess' vectors. The algorithms treat the linear transformation of input vectors as a black box that is consistent with the requirements of a Hermitian Operator. `QuantumMASALA` implements the Hamiltonian as a `class KSHam` instance that contains the `h_psi()` method which returns the $\hat{H}_{KS}(\mathbf{k})|\psi_{\mathbf{k}}\rangle$ for an input trial vector $|\psi_{\mathbf{k}}\rangle$.

The implemented Davidson Method can operate on either NumPy (CPU) and CuPy (GPU) arrays. This is possible due to the close interoperability between the two libraries.

The SCF iteration requires mixing of charge densities to accelerate convergence. `QuantumMASALA` contains multiple mixing methods, the default being Broyden mixing (Mixed [44] and General [45]). As these algorithms uses the charge densities computed from previous iterations, these routines are implemented as classes with its own memory and a method for generating the charge density for the next iteration.

When running across multiple processes via `mpiexec`, `QuantumMASALA` implements two levels of parallelism. The first involves the distribution of input **k**-points across different process groups (k-group). This is effective when working with large sets of **k**-points and scales linearly. The second distributes the `h_psi()` calls across processes in the k-group, effectively parallelizing across bands. This is useful when dealing with supercells which result a large number of occupied states. Parallelism through OpenMP is not controlled by `QuantumMASALA` and instead relies on environmental variables that specify number of threads the linked BLAS and FFT Libraries can use.

```python
from quantum_masala.lattice import RealLattice
from quantum_masala.crystal import BasisAtoms, Crystal
from quantum_masala.pseudo import UPFv2Data
from quantum_masala.gspace import GSpace
from quantum_masala.kpts import KList

from quantum_masala.dft import scf
from quantum_masala.dft.utils import printers

from quantum_masala.constants import RYDBERG
from quantum_masala.config import qtmconfig
from quantum_masala.logger import qtmlogger

world_comm = qtmconfig.get_world_comm()

# Lattice
reallat = RealLattice.from_alat(alat=5.1070,  # Bohr
                                a1=[ 0.5,  0.5,  0.5],
                                a2=[-0.5,  0.5,  0.5],
                                a3=[-0.5, -0.5,  0.5])

# Atom Basis
fe_oncv = UPFv2Data.from_file('fe', 'Fe_ONCV_PBE-1.2.upf')
fe_atoms = BasisAtoms.from_cart(
    'fe', 55.487, fe_oncv, reallat, [0., 0., 0.]
)
```

```python
crystal = Crystal(reallat, [fe_atoms, ])  # Represents the crystal

printers.print_crystal_info(crystal)

# Generating k-points from a Monkhorst Pack grid
# and reducing it to the crystal's IBZ
mpgrid_shape = (8, 8, 8)
mpgrid_shift = (True, True, True)
kpts = KList.mpgrid(crystal, mpgrid_shape, mpgrid_shift)

printers.print_kpoints(kpts)

# -----Setting up G-Space of calculation-----
ecut_wfn = 40 * RYDBERG
# NOTE: In future version, hard grid (charge/pot)
# and smooth-grid (wavefun) can be set independently
ecut_rho = 4 * ecut_wfn
grho = GSpace(crystal.recilat, ecut_rho)
gwfn = grho

printers.print_gspc_info(grho, gwfn)

# -----Spin-polarized (collinear) calculation-----
is_spin, is_noncolin = True, False
# Starting with asymmetric spin distribution else convergence
# may yield only non-magnetized states
mag_start = [0.1]
numbnd = 12  # Ensure adequate # of bands if system is not an insulator

# Control parameters that specify how occupation of eigenstates
# are computed
occ = 'smear'
smear_typ = 'gauss'
e_temp = 1E-2 * RYDBERG

# Diagonalization thresholds
conv_thr = 1E-8 * RYDBERG
diago_thr_init = 1E-2 * RYDBERG

# Calling the scf routine
out = scf(crystal, kpts, grho, gwfn, numbnd,
          is_spin, is_noncolin, mag_start=mag_start,
          occ=occ, smear_typ=smear_typ, e_temp=e_temp,
          conv_thr=conv_thr, diago_thr_init=diago_thr_init,
          iter_printer=printers.print_scf_status)

# Unpacking the output of scf function
scf_converged, rho, l_wfn_kgrp, en = out

if world_comm.rank == 0:
    print(f"SCF Routine has {'NOT' if not scf_converged else ''} "
          f"achieved convergence")
    print()

# Printing info of computed KS eigenstates
```

```
printers.print_bands(l_wfn_kgrp)

# Printing logging info
if world_comm.rankk == 0:
    print("SCF Routine has exited")
    print(qtmlogger)
```

### 5.3 'tddft' module:

The `quantum_masala.tddft_gamma` module implements propagation of the TDKS equations, which enables solving for the optical spectrum of the system. The module is designed for molecular systems, where the molecule in question is placed in a box large enough to isolate it from other molecules in neighbouring unit cells in the crystal. The calculations are limited to the gamma point $\Gamma$ of the reciprocal lattice.

Propagating the TDKS equation involves:

1. Constructing an accurate approximation to $\hat{U}(t + \Delta t, t)$, which usually involves implementing the exponentiation of the Hamiltonian

2. An accurate scheme to solve the TDKS equation and propagate the states from $t$ to $t + \Delta t$

In `QuantumMASALA`, the TDDFT module is split into 3 major sections:

- `expoper` submodule: implements the exponentiation of the Hamiltonian $\exp\{-i\Delta t\hat{H}\}$ by extending the `KSHam` class in DFT module. Currently, two implementations are available:

    - Taylor Series Expansion: truncated to order $k$.

$$\exp\{-i\Delta t\hat{H}\} \approx \sum_{r=0}^{k} \frac{(-i\Delta t)^r}{r!} \hat{H}^r \tag{52}$$

    - Split-Operator

$$\exp\{-i\Delta t\hat{H}\} \approx \exp\left\{\frac{-i\Delta t}{2}\hat{T}\right\} \exp\{-i\Delta t\hat{V}\} \exp\left\{\frac{-i\Delta t}{2}\hat{T}\right\} \tag{53}$$

Like `KSHam`, the operators here are not explicitly constructed as a dense matrix, but implemented as a routine, named `prop_psi()`, that returns the transformed vectors of input wavefunctions.

- `prop` submodule: contains implementations that use the propagators defined in `expoper` submodule and solves the TDKS equation for a given time step $t \rightarrow t + \Delta t$. Currently, this submodule contains

    - Enforced Time-Reversal Symmetry (ETRS) Propagator:

$$\bar{\psi}_{n+1} = \exp\{-i\delta t\hat{H}[\psi_n]\}\psi_n \tag{54}$$

$$\psi_{n+1} = \exp\{-i\delta t\hat{H}[\bar{\psi}_{n+1}]/2\} \exp\{-i\delta t\hat{H}[\psi_n]/2\}\psi_n \tag{55}$$

    - Split-Operator Propagator: Application of the split-operator exponential. Not time-reversal symmetric.

- `propagate` routine: combines the above two parts and solves the TDKS equations for the given time period. The propagation method used is specified by `config.tddft_exp_method` and `config.tddft_prop_method`. At each time step, a user-specified callback function is executed where quantities like dipole moment can be computed from the time-varying charge density and wavefunctions.

Using the above, the calculation of optical absorption spectra is implemented in `optical.py`. Parallelization across bands is supported in this module. Using libraries compiled with multi-threading enabled can speed up calculation. GPU Acceleration is currently not implemented as the Python wrapper to libxc does not support CuPy arrays.

```python
# TDDFT calculation requires the system's
# ground state charge and wavefun
# So the SCF routine is performed first
# followed by optical response calculation.

# ---------- SCF Calculation -----------
from quantum_masala.lattice import RealLattice
from quantum_masala.crystal import BasisAtoms, Crystal
from quantum_masala.pseudo import UPFv2Data
from quantum_masala.gspace import GSpace
from quantum_masala.kpts import KList

from quantum_masala.dft import scf
from quantum_masala.dft.utils import printers

from quantum_masala.constants import RYDBERG
from quantum_masala.config import qtmconfig
from quantum_masala.logger import qtmlogger

world_comm = qtmconfig.get_world_comm()

reallat = RealLattice.from_alat(alat=32.0,
                                a1=[1., 0., 0.],
                                a2=[0., 1., 0.],
                                a3=[0., 0., 0.83])
c_oncv = UPFv2Data.from_file('C', 'C_ONCV_PBE-1.2.upf')
h_oncv = UPFv2Data.from_file('H', 'H_ONCV_PBE-1.2.upf')
c_atoms = BasisAtoms.from_angstrom(
    'C', 12.011, c_oncv, reallat,
    (5.633200899, 6.320861303, 5.000000000),
    (6.847051545, 8.422621957, 5.000000000),
    (8.060751351, 7.721904557, 5.000000000),
    (8.060707879, 6.320636665, 5.000000000),
    (6.846898786, 5.620067381, 5.000000000),
    (5.633279551, 7.722134449, 5.000000000),
)
h_atoms = BasisAtoms.from_angstrom(
    'H', 1.008, h_oncv, reallat,
    (6.847254360, 9.512254789, 5.000000000),
    (9.004364510, 8.266639340, 5.000000000),
    (9.004297495, 5.775895755, 5.000000000),
    (6.846845929, 4.530522778, 5.000000000),
    (4.689556006, 5.776237709, 5.000000000),
    (4.689791688, 8.267023318, 5.000000000),
```

```
)
crystal = Crystal(reallat, [c_atoms, h_atoms])

kpts = KList.gamma(crystal)
ecut_wfn = 25 * RYDBERG
ecut_rho = 4 * ecut_wfn
gspc_rho = GSpace(crystal.recilat, ecut_rho)
gspc_wfn = gspc_rho

# printers.print_crystal_info(crystal)
# printers.print_kpoints(kpts)
# printers.print_gspc_info(gspc_rho, gspc_wfn)

is_spin, is_noncolin = False, False
numbnd = crystal.numel // 2
occ = 'fixed'
conv_thr = 1E-10 * RYDBERG
diago_thr_init = 1E-2 * RYDBERG

out = scf(crystal, kpts, gspc_rho, gspc_wfn,
          numbnd, is_spin, is_noncolin,
          occ=occ, conv_thr=conv_thr, diago_thr_init=diago_thr_init,
          iter_printer=printers.print_scf_status)

scf_converged, rho_scf, l_wfn_kgrp, en = out
wfn_gamma = l_wfn_kgrp[0]

# if world_comm.rank == 0:
#     print(f"SCF Routine has {'NOT' if not scf_converged else ''} "
#           f"achieved convergence")
#     print()
#
# printers.print_bands(l_wfn_kgrp)

# if world_comm.world_rank == 0:
#     print("SCF Routine has exited")
#     print(qtmlogger)

# ---------- Determining Optical Response ----------
from quantum_masala.tddft_gamma.optical import dipole_response

time_step = 0.05  # Length of time step
numsteps = 10000  # Total number of time steps
gamma = 1E-4  # Strength of delta perturbation along z-Axis at t=0

# Computing the real-time dipole response
# to a delta perturbation at t=0 along z-Axis
dip_z = dipole_response(crystal, rho_scf, wfn_gamma,
                        time_step, numsteps, gamma, 'z')

# Transforming the response to energy spectrum
en_start = 0
en_end = 40 * RYDBERG
en_step = 0.01 * RYDBERG
```

```
damp_func = 'gauss'
dip_en = dipole_spectrum(
    dip_z, time_step,
    en_start, en_end, en_step,
    damp_func
)

# Saving to file for post-processing (plotting)
import numpy as np
fname = 'dipz.npy'
with open(fname, 'wb') as f:
    np.save(f, dip_z)
```

## 5.4  'gw' module:

The implementation of GW Approximation in the package is based on BerkeleyGW [4]. Static-COHSEX and Hybertsen-Louie Plasmon Pole methods of self-energy calculation are currently available in the gw module. As of now, the module supports gapped systems. Further, *Static-Remainder* method described in [24] has been implemented to reduce number of unoccupied bands required for the convergence of Plasmon-Pole self energy values. The three primary classes in this module are: Epsilon, Sigma, and Vcoul. The classes support BerkeleyGW's input files, namely WFN[q].h5, RHO, vxc[0].dat, epsilon.inp and sigma.inp.

Input parameters for GW calculations are handled by EpsInp and SigmaInp classes. The input data can be provided either manually, by constructing the EpsInp object, or by reading BerkeleyGW-compatible input file epsilon.inp.

```
from quantum_masala.gw.io_bgw.epsinp import Epsinp

# Constructing input object manually
epsinp = Epsinp(epsilon_cutoff=10.0,
                use_wfn_hdf5=True,
                number_bands=8,
                write_vcoul=True,
                qpts=[[0.0, 0.0, 0.001],
                      [0.0, 0.0, 0.2]],
                is_q0=[True, False])

# Reading from epsilon.inp file
epsinp = Epsinp.from_epsilon_inp(filename=dirname+'epsilon.inp')

# There is an analogous system for reading SigmaInp
from quantum_masala.gw.io_bgw.sigmainp import Sigmainp
sigmainp = Sigmainp.from_sigma_inp(filename=dirname+'sigma.inp')
```

Wavefunction data generated by mean-field codes can be read using the wfn2py utility, which assumes that the incoming data satisfies BerkeleyGW's wfn.h5 specification. The data is stored as a NamedTuple object. Alternatively, the data generated from scf runs within QuantumMASALA can also be passed directly to Epsilon and Sigma constructors in the form of quantum_masala objects as shown later.

We also require wavefunctions on a shifted grid to calculate dielectric matrix at $q \rightarrow 0$. This shifted k-grid dataset will be referred to as wfnqdata.

```
from quantum_masala.gw.io_bgw.wfn2py import wfn2py
wfndata = wfn2py(dirname+'WFN.h5')
wfnqdata = wfn2py(dirname+'WFNq.h5')
```

### 5.4.1 Class Epsilon

Calculates RPA irreducible polarizability matrix $P$, and inverse dielectric matrix $\epsilon^{-1}$ using DFT energy eigenfunctions.

Epsilon class can be initialized by either directly passing the required quantum_masala objects or by passing the input objects discussed earlier.

```
from quantum_masala.gw.epsilon import Epsilon

# Initializing using data objects
epsilon = Epsilon.from_data(wfndata=wfndata,
                            wfnqdata=wfnqdata,
                            epsinp=epsinp)

# Alternative, manual initialization
epsilon = Epsilon(
    crystal = wfndata.crystal,
    gspace = wfndata.grho,
    kpts = wfndata.kpts,
    kptsq = wfnqdata.kpts,
    l_wfn = wfndata.l_wfn,
    l_wfnq = wfnqdata.l_wfn,
    l_gsp_wfn = wfndata.l_gk,
    l_gsp_wfnq = wfnqdata.l_gk,
    epsinp = epsinp,
    qpts = QPoints.from_cryst(recilat = wfndata.kpts.recilat,
                              is_q0 = epsinp.is_q0,
                              cryst = *epsinp.qpts))
```

The three main steps involved in the calculation have been mapped to corresponding functions:

1. matrix_elements: Calculation of plane wave matrix elements

$$M_{nn'}(\mathbf{k}, \mathbf{q}, \mathbf{G}) = \langle n\,\mathbf{k+q}| \, e^{i(\mathbf{q+G})\cdot\mathbf{r}} \, |n'\,\mathbf{k}\rangle \tag{56}$$

where the **G**-vectors included in the calculation satisfy $|\mathbf{q} + \mathbf{G}|^2 < E_{\text{cut}}$. Since this is a convolution in k-space, the time complexity can be reduced from $\mathcal{O}\left(N_G^2\right)$ to $\mathcal{O}\left(N_G \ln N_G\right)$ by using Fast Fourier Transform, where $N_G$ the number of reciprocal lattice vectors in the wavefunction.

$$M_{nn'}(\mathbf{k}, \mathbf{q}, \{\mathbf{G}\}) = \text{FFT}^{-1}\left(\phi^*_{n,\mathbf{k+q}}(\mathbf{r})\phi_{n',\mathbf{k}}(\mathbf{r})\right). \tag{57}$$

where $\phi_{n',\mathbf{k}}(\mathbf{r}) = \text{FFT}\left(\psi_{n\mathbf{k}}(\mathbf{G})\right)$.

2. polarizability: Calculation of RPA polarizability matrix $P$

$$P_{\mathbf{GG'}}(\mathbf{q}; 0) = \sum_{n}^{\text{occ}} \sum_{n'}^{\text{emp}} \sum_{\mathbf{k}} \frac{\langle n'\mathbf{k}| \, e^{-i(\mathbf{q+G})\cdot\mathbf{r}} \, |n\mathbf{k+q}\rangle \, \langle n\mathbf{k+q}| \, e^{i(\mathbf{q+G'})\cdot\mathbf{r}} \, |n'\mathbf{k}\rangle}{E_{n\mathbf{k+q}} - E_{n'\mathbf{k}}}. \tag{58}$$

3. `epsilon_inverse`: Calculation of (static) epsilon-inverse matrix

$$\epsilon_{\mathbf{GG'}}(\mathbf{q}) = \delta_{\mathbf{GG'}} - v(\mathbf{q+G})\,P_{\mathbf{GG'}}(\mathbf{q}) \tag{59}$$

where $v(\mathbf{q}+\mathbf{G}) = \frac{8\pi}{|\mathbf{q}+\mathbf{G}|^2}$ is bare Coulomb potential, written in Rydberg units. If this formula is used as-is for the case where $|\mathbf{q}| = |\mathbf{G}| = 0$, the resulting $v(\mathbf{q=0}, \mathbf{G=0})$ blows up as $1/q^2$. However, for 3D gapped systems, the matrix elements $\left| M_{nn'}(\mathbf{k}, \mathbf{q} \to \mathbf{0}, \mathbf{G=0}) \right| \sim q$ cancel the Coulomb divergence and $\epsilon_{00}(\mathbf{q} \to \mathbf{0}) \sim q^2/q^2$ which is a finite number. In order to calculate $\epsilon_{00}(\mathbf{q} = \mathbf{0})$, we use the scheme specified in [4], wherein $q = 0$ is replaced with a small but non-zero value. Since matrix element calculation involves the eigenvectors $|n\mathbf{k+q}\rangle$, having a non-$\Gamma$-centered $\mathbf{q} \to 0$ point requires mean-field eigenvectors over a shifted $k$-grid.

```python
# Calculate plane wave matrix elements
mtxel = next(epsilon.matrix_elements(i_q=i_q))

# Calculate polarizability matrix
chimat = epsilon.polarizability(mtxel)

# Calculate epsilon inverse matrix
epsinv = epsilon.epsilon_inverse(i_q = i_q,
                                 polarizability_matrix = chimat,
                                 store = True)
```

### 5.4.2 Class Sigma

The `Sigma` class provides methods for calculation of GW self-energy. The self-energy approximations available in the module are: Hartree-Fock bare exchange, static-COHSEX [3] and Hybertsen-Louie Plasmon Pole method [16] [23].

```python
from quantum_masala.gw.sigma import Sigma

sigma = Sigma.from_data(
    wfndata=wfndata,
    wfnqdata=wfnqdata,
    sigmainp=sigmainp,
    epsinp=epsinp,
    l_epsmats=epsilon.l_epsinv,
    rho=rho,
    vxc=vxc,
    outdir=dirname+"temp/")
```

Conventionally, self-energies are divided into a frequency independent Fock exchange energy term $\Sigma_{\mathrm{X}}$, and a correlation term $\Sigma_{\mathrm{Corr}}$, which is further divided into a screened exchange (minus bare exchange) part $\Sigma_{\mathrm{SX}} \equiv \Sigma_{\mathrm{SEX}} - \Sigma_{\mathrm{X}}$ and the Coulomb-hole part $\Sigma_{\mathrm{COH}}$:

$$\mathrm{Re}(\Sigma) = \Sigma_{\mathrm{X}} + \Sigma_{\mathrm{SX}} + \Sigma_{\mathrm{COH}}$$

Following are the core methods provided by `Sigma` class:

- Hartree-Fock bare exchange : `sigma_x`

- Static COHSEX

    - Screened exchange : `sigma_sx_static`

– Coulomb hole partial sum : `sigma_ch_static`

– Coulomb hole exact : `sigma_ch_static_exact`

- Hybertsen Louie Plasmon Pole

    – Screened exchange : `sigma_sx_gpp`

    – Coulomb hole partial sum : `sigma_ch_gpp`

- A `matrix_elements` function, analogous to that in `Epsilon` class.

```
# Do a GPP quasiparticle energy calculation
sigma.calculate_gpp()

# Do a static COHSEX quasiparticle energy calculation
sigma.calculate_static_cohsex()

# Individual function calls
sigma.sigma_sx_static()
```

### 5.4.3 Class Vcoul

The `Vcoul` class provides methods to calculate bare coulomb interaction, $v(\mathbf{q}, \mathbf{G}) = 8\pi/|\mathbf{q}+\mathbf{G}|^2$, along with methods required to average the potential over mini-Brillouin zones.

Coulomb potential $\sim 1/|\mathbf{q}+\mathbf{G}|^2$ diverges when $q = G = 0$. However, the Coulomb potential values over a discretized $G$-space grid only represent the average potential in the mini-Brillouin zones corresponding to those grid points. As a result we construct,

$$V_{\text{grid}}(\mathbf{q}, \mathbf{G}) = \int_{\text{minibz}} \frac{2}{|\mathbf{q}+\mathbf{G}|^2} \mathrm{d}^3\mathbf{G}$$

(in Rydberg units) which turns out to be convergent for all grid points. Even though the apparent divergence occurs only for the $q = G = 0$ case, the averaging is typically performed for all q-points and all G-points for better accuracy [46].

Here we list the primary methods in `Vcoul` class:

- `v_bare`: Bare Coulomb potential for a given q-point.

- `v_minibz_montecarlo`: Averaged bare Coulomb potential, where naive Monte Carlo averaging has been used.

- `v_minibz_sphere`: Calculates exact integral of Coulomb potential over a spherical region with the same volume as the corresponding mini-Brillouin zone.

- `v_minibz_montecarlo_hybrid`: Calculates exact integral within a sphere inscribed in the mini-Brillouin zone and uses Monte Carlo averaging for the rest.

```
# Initialization
vcoul = Vcoul( gspace = wfndata.grho,
               qpts = qpts,
               bare_coulomb_cutoff = epsinv.epsilon_cutoff)

# Populate with coulomb potential averaged within mini-Brillouin zones
vcoul.calculate_vcoul(averaging_func=vcoul.v_minibz_montecarlo_hybrid)

# Print vcoul in BerkeleyGW format
```

```
vcoul.write_vcoul()

# Load vcoul from BerkeleyGW format
vcoul.load_vcoul(filename="vcoul.dat")
```

## 6   Results

In this section, we demonstrate the current capabilities of `QuantumMASALA` by showing some results obtained from it. Here, we present benchmarking results that show the accuracy of the implemented calculations in the code. We also compare its consistency and performances with standard packages like `QuantumESPRESSO` for DFT and TDDFT, and BerkeleyGW for GW. This section has been split on the basis of the calculation method/routine where both the accuracy and the performance of the code is discussed.

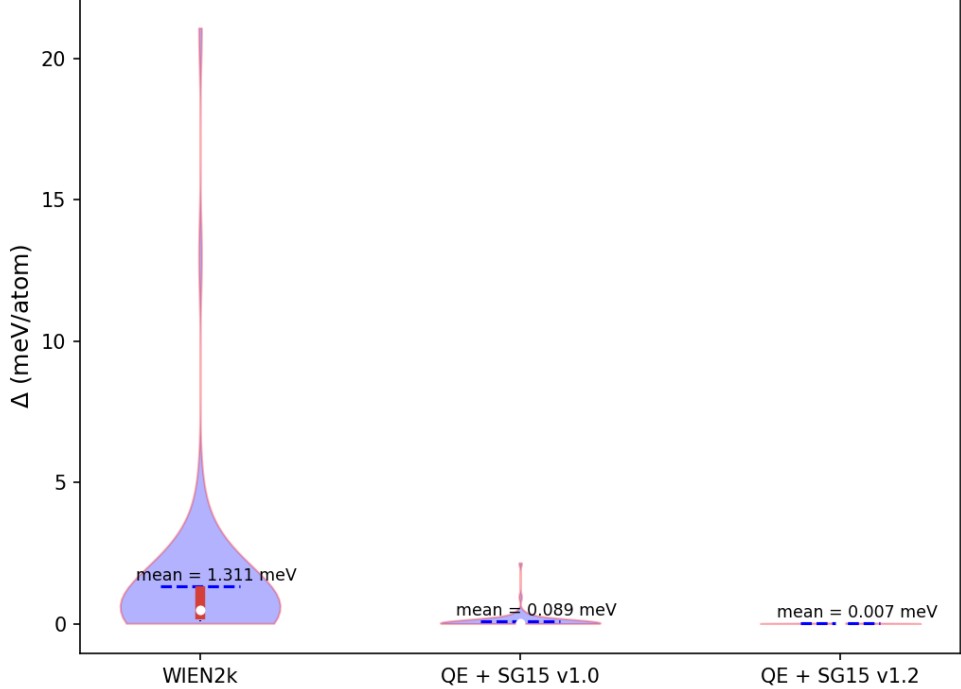

Figure 2: Distribution of elementwise Δ values computed between the following pair of DFT codes: `QuantumMASALA` vs `WIEN2k`, `QuantumMASALA` vs `QuantumESPRESSO` (with SG15 v1.0) and `QuantumMASALA` vs `QuantumESPRESSO` (with SG15 v1.2)

### 6.1   DFT

For DFT calculations, the Δ Benchmark [42] provides a standardized protocol that quantifies the differences in results obtained between different solid state codes. The Δ value is the root-mean-square difference between the equation of state computed from two codes, averaged over a benchmark set of 71 elemental crystals. The equation of state is determined by fitting

| $k$-pt. | LDA | QuantumMASALA | | BerkeleyGW | | Exp. |
| | | Static | GPP | Static | GPP | |
|---|---|---|---|---|---|---|
| $\Gamma_{1v}$ | −11.93 | −12.75 | −11.66 | −12.75 | −11.66 | −12.50 |
| $\Gamma'_{25v}$ | 0.00 | 0.00 | 0.00 | 0.00 | 0.00 | 0.00 |
| $\Gamma_{15c}$ | 2.56 | 3.87 | 3.36 | 3.87 | 3.36 | 3.34 |
| $\Gamma'_{2c}$ | 3.26 | 4.32 | 3.94 | 4.32 | 3.94 | 4.18 |
| $X_{1v}$ | −7.78 | −8.31 | −7.76 | −8.31 | −7.76 | |
| $X_{4v}$ | −2.86 | −2.86 | −2.85 | −2.86 | −2.85 | −2.90 |
| $X_{1c}$ | 0.66 | 2.07 | 1.49 | 2.07 | 1.49 | 1.30 |
| $L'_{2v}$ | −9.58 | −10.28 | −9.48 | −10.28 | −9.48 | −9.30 |
| $L_{1v}$ | −6.98 | −7.24 | −6.92 | −7.24 | −6.92 | −6.80 |
| $L'_{3v}$ | −1.20 | −1.20 | −1.21 | −1.20 | −1.21 | −1.20 |
| $L_{1c}$ | 1.49 | 2.64 | 2.22 | 2.64 | 2.22 | 2.04 |
| $L_{3c}$ | 3.33 | 4.80 | 4.23 | 4.80 | 4.23 | 3.90 |

*Note:* All values are in *eV*.

Table 1: Comparison of quasiparticle energies for Si, calculated using `QuantumMASALA` and `BerkeleyGW`. Experimental data taken from [47]

the Birch-Murnaghan Equation of state [48] using energies computed across different unit-cell volumes of the crystal.

The $\Delta$ Benchmark has been run on `QuantumMASALA` with the SG15 [43] set of ONCV pseudopotentials (ver 1.2). The generated Birch-Murnaghan parameters can be found on the Appendix section for reference. All electron DFT calculations are considered to be the standard as they are not affected by the pseudization of potential. The $\Delta$-value between `QuantumMASALA` and `WIEN2k` [49, 50], an all-electron DFT code is **1.3 meV/atom**, which matches with `QuantumESPRESSO` + SG15 pseudopotential. This demonstrates the accuracy of the DFT routines implemmented in `QuantumMASALA`.

## 6.2  GW

For GWA calculations, we present a comparison of quasiparticle energy calculation results from `BerkeleyGW` and from `QuantumMASALA` for Static COHSEX, and Hybertsen-Louie Plasmon Pole with Static Remainder correction. The convergence study presented in [47] was used to decide parameters required for convergence of Silicon quasiparticle energies for $6 \times 6 \times 6$ k-grid within 10 meV. For the mean field calculation (using DFT), the wave function was expanded in plane waves with energy upto 25 Ry. A dielectric cut off of 25 Ry was used. 274 empty bands were included in CH summation. The calculated quasiparticle energies obtained using `QuantumMASALA` were found to match the results obtained using `BerkeleyGW` to within 100 $\mu$eV.

## 6.3  TDDFT

The accuracy of the TDDFT routines in `QuantumMASALA` is assessed by evaluating the dynamical polarizability matrix elements $\chi_{ij}(\omega)$, whose trace is proportional to the optical absorption spectrum $\sigma(\omega)$. This is validated with the same computed from `QuantumESPRESSO`'s TDDFPT module [51, 52]. Although, the implementations are not identical, it is expected to give similar results. Below we present the results obtained for the Methane Molecule. Fig. (3)

shows the comparison between the dynamical polarizability between `QuantumMASALA` and `QuantumESPRESSO`. From the figure, it is clear that the results agree well with each other. We plan to implement improved algorithms to enable the package to target bigger systems in the future.

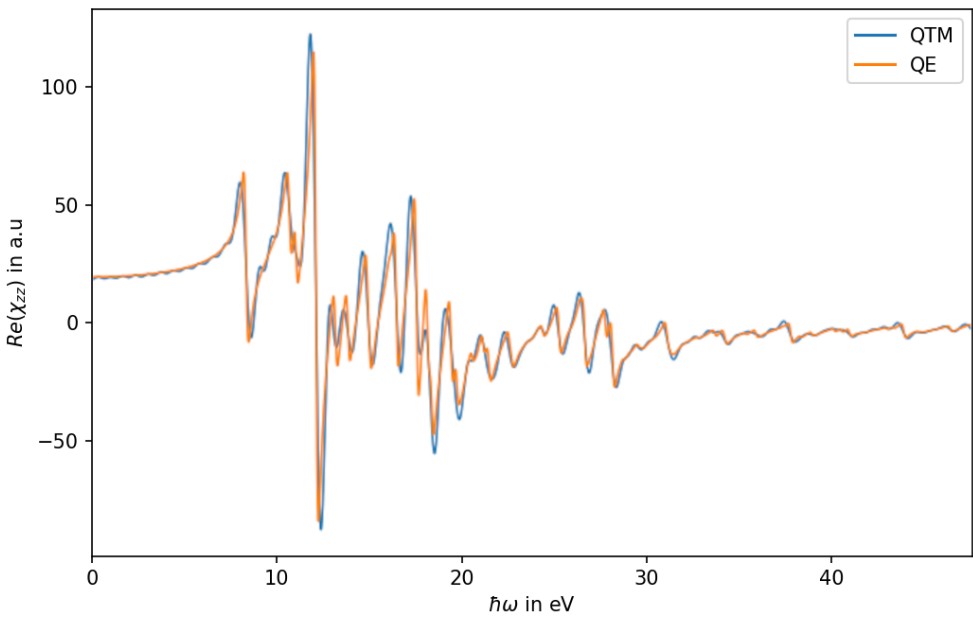

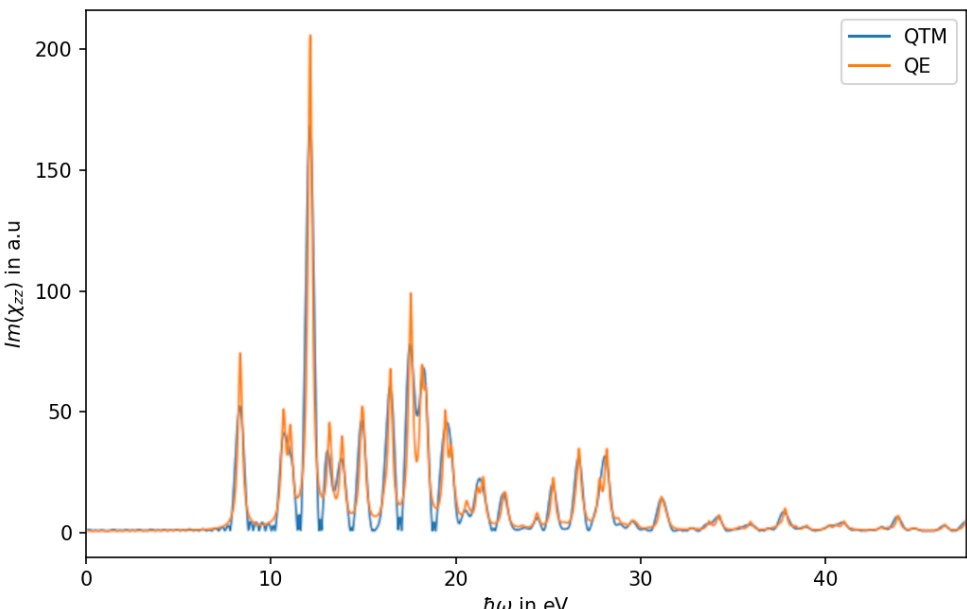

Figure 3: Polarizability Spectra of Methane molecule obtained from `QuantumESPRESSO`'s TDDFPT package (orange) and the TDDFT implementation in `QuantumMASALA` (blue)

## 6.4 Performance

Although `QuantumMASALA` is primarily designed to be compact and simple, the performance-critical sections of the code have been extensively optimized through efficient utilisation of linear algebra routines provided by NumPy+Scipy(CPU)/CuPy(GPU). This allows the code to minimize the performance gap to its compiled counterparts. An ideal implementation is expected to maximize the fraction of runtime spent in library routines instead of the Python interpreter.

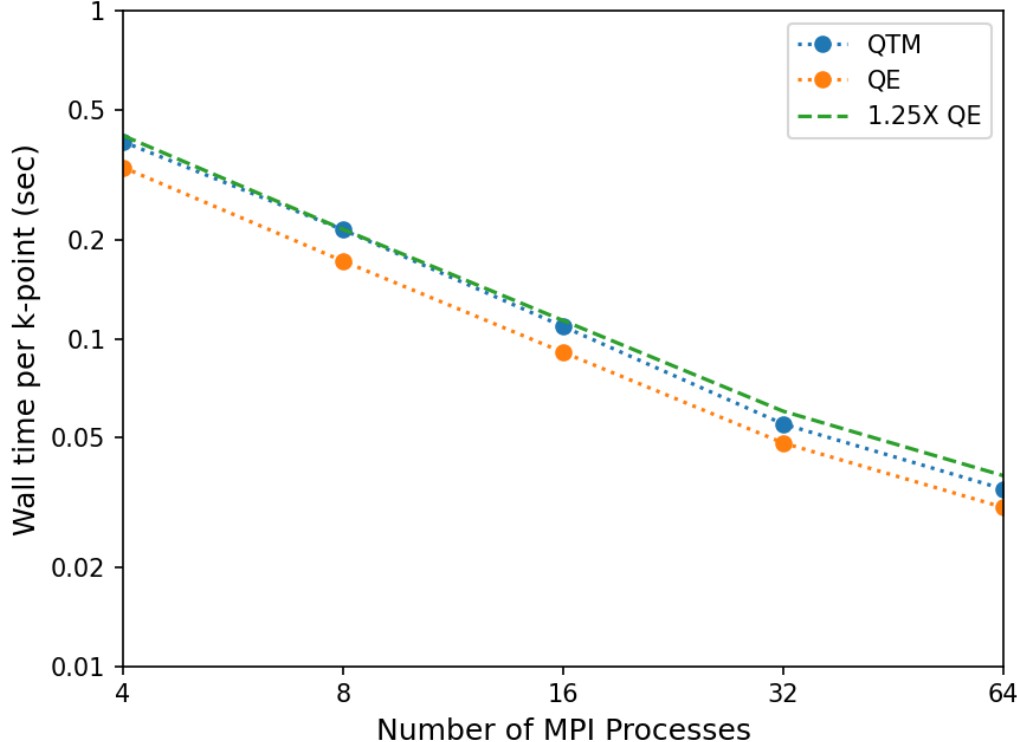

Figure 4: Wall time vs # of processes used for SCF calculation of the Fe crystal in the $\Delta$ Benchmark with k-point parallelization only. Blue (Orange) points correspond to wall times for `QuantumMASALA` (`QuantumESPRESSO`). The green dashed line corresponds to wall times which are 25% slower than `QuantumESPRESSO`. The dashed lines are guides to the eye.

To assess the performance of `QuantumMASALA`, we have measured the runtimes of DFT calculations which were performed with varying number of processors. The wall times were compared against the same performed on an installatiton of `QuantumESPRESSO` v7.1 that is linked to the same libraries/dependencies as `QuantumMASALA` (OpenBLAS 0.3.20, FFTW 3.3.10, MPICH 4.0.2). The benchmarks were performed in a workstation equipped with 2X AMD EPYC 7532 32-Core Processors, around 1 Terabyte of DDR4 2666MHz RAM. It must be noted that multithreading is disabled in our benchmarks.

We present results from two benchmarks, each targeting a particular parallelization mode. The first, a crystalline Iron system used in delta benchmark. In this benchmark, the k-points for sampling the Brillouin zone are distributed among MPI tasks. Fig. (4) shows the reduction in the wall time taken as the number of processors (same as MPI tasks) are increased. It also shows the timings for the same calculation using the k-point parallelization in `QuantumESPRESSO`.

The second benchmark is that of supercell of Silicon containing 6×6×6 silicon unit-cells (432 atoms), with only the Γ point sampling of the Brillouin zone using band parallelization. We note, that while the bands distributed among various MPI tasks, the final diagonalization of the subspace projected matrix is performed serially. Fig. (5) shows the wall time as a function of MPI tasks. We also compare the wall times to `QuantumESPRESSO` for the same calculation using identical parallelization scheme (and serial diagonalization). Figs. (4) and (5) show that `QuantumMASALA` can tackle medium-scale systems with performance comparable to `QuantumESPRESSO`. This enables implementations built using this code to not be bottle necked by the relatively-slow programming language while leveraging its simplicity for rapid prototyping.

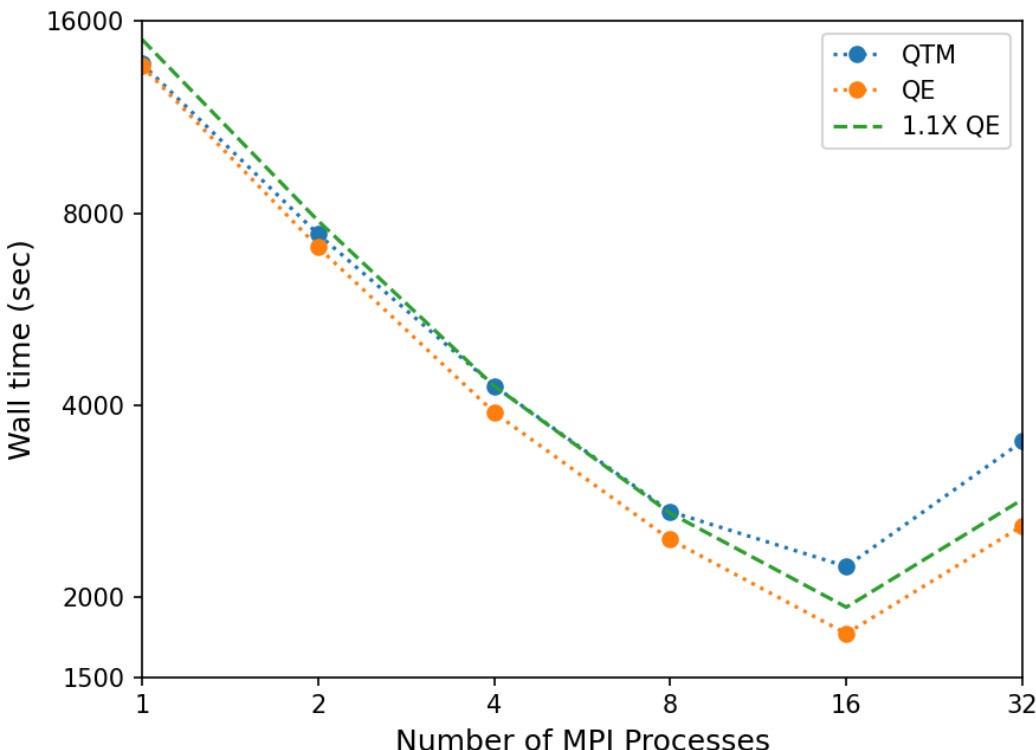

Figure 5: Wall time vs # of processes used for SCF calculation of a $6 \times 6 \times 6$ super-cell of silicon (Γ-point only) with parallelization across bands. Blue (Orange) points correspond to wall times for `QuantumMASALA` (`QuantumESPRESSO`). The green dashed line corresponds to wall times which are 10% slower than `QuantumESPRESSO`. The dashed lines are guides to the eye.

# 7 Conclusion

We have presented `QuantumMASALA`, a compact package that implements different electronic-structure methods in Python. Within just 8000 lines of pure Python code, we have implemented Density Functional Theory (DFT), Time-dependent Density Functional Theory (TD-DFT) and the GW Method for periodic systems. We have demonstrated the ability of the code to scale across multiple process cores and to run in Graphical Processing Units (GPU) with

the help of easily-accessible Python libraries. With `QuantumESPRESSO` and `BerkeleyGW` I/O interfaces implemented, it can also be used as a substitute for small scale calculations. `QuantumMASALA` is extremely light and simple, making it an ideal tool for teaching and learning *ab initio* methods. Further, its performance and modularity makes it a perfect framework for developing and testing new methods for *ab initio* electronic structure calculations.

# 8 Appendix

## 8.1 $\Delta$ Benchmark: Complete Results

Table 2: Result of the $\Delta$ Benchmark of `QuantumMASALA`: Equation of State (EOS) and $\Delta$-value with respect to `WIEN2k`

|     | $Z_{val}$ | k-mesh | $V_0$ [Å/at] | $B_0$ [GPa] | $B_1$ [-] | $\Delta_{\text{WIEN2k}}$ [meV/at] | $\Delta_{\text{QE,1.0}}$ [meV/at] | $\Delta_{\text{QE,1.2}}$ [meV/at] |
|-----|-----|--------|--------|---------|-------|---------|---------|---------|
| H   | 1   | 28×28×20 | 17.356 | 10.287  | 2.660 | 0.072  | 0     | 0     |
| He  | 2   | 40×40×22 | 17.711 | 0.880   | 6.202 | 0.013  | 0     | 0     |
| Li  | 3   | 38×38×38 | 20.244 | 13.846  | 3.342 | 0.074  | 0.001 | 0.001 |
| Be  | 4   | 52×52×28 | 7.931  | 123.704 | 3.306 | 0.600  | 0.001 | 0     |
| B   | 3   | 26×26×24 | 7.180  | 235.356 | 3.425 | 3.132  | 0.003 | 0     |
| C   | 4   | 48×48×12 | 11.588 | 207.053 | 3.589 | 2.266  | 0.052 | 0.001 |
| N   | 5   | 16×16×16 | 28.769 | 53.412  | 3.599 | 1.354  | 0.052 | 0.047 |
| O   | 6   | 26×24×24 | 18.597 | 50.514  | 3.918 | 0.378  | 0.05  | 0.013 |
| F   | 7   | 16×28×14 | 19.283 | 33.909  | 3.941 | 0.866  | 0.015 | 0.015 |
| Ne  | 8   | 22×22×22 | 24.259 | 1.352   | 7.217 | 0.021  | 0     | 0     |
| Na  | 9   | 32×32×32 | 37.113 | 7.761   | 3.698 | 0.587  | 0.001 | 0     |
| Mg  | 10  | 36×36×20 | 22.960 | 36.553  | 4.066 | 0.221  | 0.001 | 0.002 |
| Al  | 11  | 24×24×24 | 16.529 | 77.749  | 4.870 | 0.858  | 0.012 | 0.007 |
| Si  | 4   | 32×32×32 | 20.543 | 87.440  | 4.264 | 1.313  | 0.387 | 0.001 |
| P   | 5   | 30×8×22  | 21.475 | 67.802  | 4.311 | 0.049  | 0.028 | 0     |
| S   | 6   | 38×38×38 | 17.293 | 83.955  | 4.101 | 1.082  | 0.944 | 0.003 |
| Cl  | 7   | 12×24×12 | 39.328 | 18.707  | 4.378 | 2.044  | 0.236 | 0     |
| Ar  | 8   | 16×16×16 | 52.484 | 0.753   | 7.705 | 0.009  | 0.028 | 0     |
| K   | 9   | 20×20×20 | 73.655 | 3.601   | 4.001 | 0.032  | 0.001 | 0.001 |
| Ca  | 10  | 18×18×18 | 42.172 | 17.608  | 3.252 | 0.136  | 0.009 | 0.01  |
| Sc  | 11  | 34×34×20 | 24.643 | 54.575  | 3.354 | 0.269  | 0.009 | 0.007 |
| Ti  | 12  | 40×40×22 | 17.407 | 112.070 | 3.540 | 0.406  | 0.004 | 0.004 |
| V   | 13  | 34×34×34 | 13.468 | 182.316 | 3.943 | 0.686  | 0.002 | 0.004 |
| Cr  | 14  | 36×36×36 | 12.441 | 117.003 | 6.278 | 21.075 | 0.132 | 0.121 |
| Mn  | 15  | 28×28×28 | 11.899 | 128.152 | 4.530 | 12.988 | 0.063 | 0.001 |
| Fe  | 16  | 36×36×36 | 11.454 | 180.311 | 6.856 | 4.665  | 0.074 | 0     |
| Co  | 17  | 46×46×24 | 10.923 | 211.271 | 4.761 | 2.830  | 0.011 | 0.001 |
| Ni  | 18  | 28×28×28 | 10.943 | 195.429 | 5.074 | 2.215  | 0.003 | 0     |
| Cu  | 19  | 28×28×28 | 11.986 | 138.974 | 5.101 | 0.916  | 0.003 | 0.004 |
| Zn  | 20  | 44×44×20 | 15.172 | 75.751  | 5.342 | 0.359  | 0.004 | 0.001 |
| Ga  | 13  | 22×12×22 | 20.339 | 49.764  | 6.223 | 0.354  | 0.141 | 0.001 |
| Ge  | 14  | 30×30×30 | 23.981 | 58.972  | 4.888 | 0.845  | 0.003 | 0.002 |
| As  | 5   | 30×30×10 | 22.673 | 68.063  | 4.248 | 1.491  | 0.235 | 0.001 |

|     | $Z_{val}$ | k-mesh | $V_0$ [Å/at] | $B_0$ [GPa] | $B_1$ [-] | $\Delta_{\text{WIEN2k}}$ [meV/at] | $\Delta_{\text{QE,1.0}}$ [meV/at] | $\Delta_{\text{QE,1.2}}$ [meV/at] |
|-----|-----------|--------|--------------|-------------|-----------|------------------|------------------|------------------|
| Se  | 6   | 26×26×20 | 29.806  | 46.942  | 4.436 | 2.787 | 2.151 | 0.002 |
| Br  | 7   | 12×24×12 | 39.670  | 22.378  | 4.832 | 1.091 | 0     | 0     |
| Kr  | 8   | 16×16×16 | 65.984  | 0.646   | 7.193 | 0.037 | 0     | 0     |
| Rb  | 9   | 18×18×18 | 91.007  | 2.792   | 3.788 | 0.090 | 0.007 | 0.012 |
| Sr  | 10  | 16×16×16 | 54.434  | 11.223  | 4.980 | 0.213 | 0.052 | 0.064 |
| Y   | 11  | 32×32×18 | 32.859  | 41.365  | 3.157 | 0.138 | 0.008 | 0.007 |
| Zr  | 12  | 36×36×20 | 23.398  | 94.028  | 3.278 | 0.281 | 0.004 | 0.004 |
| Nb  | 13  | 30×30×30 | 18.149  | 169.968 | 3.606 | 0.428 | 0.027 | 0.013 |
| Mo  | 14  | 32×32×32 | 15.782  | 260.153 | 4.284 | 0.283 | 0.011 | 0.012 |
| Tc  | N/A | N/A      | N/A     | N/A     | N/A   | N/A   | N/A   | N/A   |
| Ru  | 16  | 42×42×24 | 13.770  | 312.134 | 4.848 | 0.489 | 0.002 | 0.001 |
| Rh  | 17  | 26×26×26 | 14.050  | 256.947 | 5.175 | 0.548 | 0.011 | 0.002 |
| Pd  | 18  | 26×26×26 | 15.311  | 169.601 | 5.548 | 0.084 | 0.014 | 0.007 |
| Ag  | 19  | 24×24×24 | 17.825  | 91.070  | 6.035 | 0.353 | 0.006 | 0.006 |
| Cd  | 20  | 38×38×18 | 22.950  | 43.607  | 6.929 | 1.097 | 0.005 | 0.005 |
| In  | 13  | 30×30×20 | 27.529  | 36.051  | 5.033 | 0.413 | 0.117 | 0.004 |
| Sn  | 14  | 26×26×26 | 36.836  | 35.725  | 4.902 | 0.502 | 0.33  | 0.002 |
| Sb  | 15  | 26×26×8  | 31.773  | 50.381  | 4.538 | 0.360 | 0.121 | 0     |
| Te  | 16  | 26×26×16 | 35.075  | 44.638  | 4.616 | 0.578 | 0.376 | 0     |
| I   | 17  | 12×22×10 | 50.398  | 18.595  | 5.046 | 0.577 | 0.093 | 0     |
| Xe  | 18  | 14×14×14 | 87.128  | 0.538   | 7.165 | 0.016 | 0.04  | 0     |
| Cs  | 9   | 16×16×16 | 116.932 | 1.960   | 3.523 | 0.051 | 0.009 | 0.02  |
| Ba  | 10  | 20×20×20 | 63.227  | 8.719   | 2.064 | 0.097 | 0.002 | 0.003 |
| Lu  | N/A | N/A      | N/A     | N/A     | N/A   | N/A   | N/A   | N/A   |
| Hf  | 26  | 36×36×20 | 22.596  | 108.614 | 3.454 | 1.511 | 0.001 | 0.002 |
| Ta  | 27  | 30×30×30 | 18.324  | 194.941 | 3.660 | 1.628 | 0.008 | 0.011 |
| W   | 28  | 32×32×32 | 16.150  | 301.099 | 4.131 | 0.638 | 0.003 | 0.001 |
| Re  | 15  | 42×42×22 | 14.940  | 365.160 | 4.411 | 1.474 | 0.021 | 0.001 |
| Os  | 16  | 42×42×24 | 14.255  | 398.955 | 4.811 | 2.209 | 0.014 | 0.001 |
| Ir  | 17  | 26×26×26 | 14.472  | 349.167 | 5.115 | 2.155 | 0.01  | 0.007 |
| Pt  | 18  | 26×26×26 | 15.687  | 246.034 | 5.476 | 2.416 | 0.011 | 0.008 |
| Au  | 19  | 24×24×24 | 17.982  | 138.738 | 6.013 | 0.276 | 0.002 | 0.004 |
| Hg  | 20  | 24×24×28 | 29.550  | 7.767   | 9.868 | 0.116 | 0.001 | 0.005 |
| Tl  | 13  | 32×32×18 | 31.361  | 26.875  | 5.419 | 0.177 | 0.003 | 0.003 |
| Pb  | 14  | 20×20×20 | 31.945  | 40.048  | 5.467 | 0.414 | 0.008 | 0.008 |
| Bi  | 15  | 26×26×8  | 36.917  | 42.818  | 4.684 | 0.129 | 0.002 | 0.002 |
| Po  | N/A | N/A      | N/A     | N/A     | N/A   | N/A   | N/A   | N/A   |
| Rn  | N/A | N/A      | N/A     | N/A     | N/A   | N/A   | N/A   | N/A   |
|     |     |          |         |         | $\langle\Delta\rangle$ | 1.298 | 0.089 | 0.006 |
|     |     |          |         |         | $\sigma_\Delta$ | 2.980 | 0.289 | 0.010 |

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
