# Peer review of "Quantum MASALA: Quantum MAterialS Ab initio eLectronic-structure pAckage"

_SciPost Physics Codebases_

## Round 2 · Referee Report · Anonymous (Referee 1) · 2024-11-4

Strengths

Present a new DFT code in Python, that can be useful for future implementations and testing.

Weaknesses

No particular weaknesses, just some point to be clarified.

Report

In this manuscript, the authors present a new DFT code written in Python: QuantumMasala. The paper is clear and well written, but I think there are some points that the authors should clarify/correct before I can suggest publication:

1) Reference 13 to the Yambo code is an old reference, please replace it with: Sangalli D, Ferretti A, Miranda H, Attaccalite C, Marri I, Cannuccia E, Melo P, Marsili M, Paleari F, Marrazzo A, Prandini G. Many-body perturbation theory calculations using the yambo code. Journal of Physics: Condensed Matter. 2019 May 29;31(32):325902.

2) In the manuscript, the authors discuss MPI and GPU parallelization, but do not mention openMP. Does the code support openMP parallelization?

3) At page 5 they say that the cutoff on V_KS is 4 times larger of the wave-function cutoff, is the same for the density?

4) The TD-DFT is not entirely clear. On page 6 they say that the perturbation is V=\delta(t) r \dot x. Have the authors implemented the TD-DFT in length gauge using the dipole <r>? This is quite complicated and requires the use of the Berry phase (see e.g. Phys. Rev. B 94, 035149) or did they use the most standard formulation in the velocity gauge with pA coupling (see Phys. Rev. B 62, 7998, 2000)? They should clarify this point and correct the equations according to the correct implementation.

5) The authors have implemented the Hybertsen-Louie model for the plasmon pole in their code. However, this model is known to overestimate the gaps and a large part of the scientific community is moving towards other models such as the Godby-Needs etc. For a discussion of the failure of the HL model see Phys. Rev. B 84, 241201, 2011. This should be mentioned in the manuscript.

6) Notation at page 16 is a bit confusion, why G vectors become q vectors at a certain point?

7) On page 35 the integration of the Coulomb potential is discussed, can the authors give more details? In general, how many Monte Carlo points are required to converge to this integral? Does the integral change in low dimensional systems?

8) Is the output of the code written in HDF5 format? Are there some utilities to analyze results?

9) How is the memory scaling of the code compared with QE?

10) Is the output of QuantumMasala compatible with many tools available as: Wannier90, PAOFLAW (https://aflow.org/src/paoflow/), BandUP, CUBE, VESTA, Bader(https://theory.cm.utexas.edu/henkelman/code/bader/),? Is it compatible with QE

Recommendation

Ask for minor revision

---

## Round 2 · Referee Report · Anonymous (Referee 2) · 2024-11-13

Strengths

interesting approach to use python for an electronic structure code

Report

The authors present a code for electronic structure calculations based on plane waves. Besides DFT, TDDFT and the GW method are implemented. It is interesting that python as a programming language has been used, which allows a relatively small size of the code.

I recommend publication with minor changes:

1) The equations should be checked, e.g.: eqn. 1 V_aux is used, but in (11) it is called V_KS. Better stick to one notation eqn. 11 < G_{m} + k| not < G_{m+k}| i.e. k should not be subscript eqn. 21 the notation is somewhat ill-conceived, as \vu is x,y,z but then r_{\nu} would be r_x,r_y,r_z instead of just x,y,z eqn. 55 on the left hand side it should be \psi_{n+1} with a bar as in eqn. 54

In general, references to the individual equations should be given. E.g. section 2.5, especially equations 30 - 45, are essentially as in [4] and [29] and use the same notation.

2) Acronyms should be explained where they appear for the first time, e.g.: HLPP, GPP ...

3) Concerning the code: in which way is symmetry used (space groups etc)?

4) Concerning the results: besides comparing numbers as e.g. in Table 1, an additional plot of the band structure would make a comparison easier and more pleasant for the reader.

Recommendation

Ask for minor revision

---

## Round 2 · Referee Report · Anonymous (Referee 3) · 2024-12-2

Strengths

1- Fairly complete DFT code in a Python environment. 2- The code is easy to install 3- The article presents the DFT method and the use of code in a very pedagogical way.

Report

This article presents a new DFT code, called Quantum MASALA, written quite compactly in Python. Density Functional Theory (DFT), Time-dependent Density Functional Theory (TD-DFT) and the GW method are implemented. The fact that it is written in Python means it can be used on multiple platforms. According to the Benchmark presented in the manuscript, the code is well optimized. It can be used in parallel (MPI) version, plus openMP thanks to the use of parallel libraries (MKL). It seems to me that this code should be very useful for those wishing to discover and use DFT while studying the method itself and its numerical implementation. I therefore recommend this manuscript for publication in SciPost Physics Codebases. Remarks: (1) Small typo: The link in the footnote of page 3 is broken. (2) A small remarks on the installation on linux (ubuntu): After following the installation procedure described in the manuscript, I got an error because I also had to install git to run a test. I am a little surprised that git is necessary. After installing git, the test worked fine but with an error at the beginning: " fatal: neither this nor any of its parent directories (up to mount point /xxx/) is a git repository Stop at file system boundary (GIT_DISCOVERY_ACROSS_FILESYSTEM not defined). Error retrieving project git info: Command '['git', 'log', '-1', '--format=%H %ad', '--date=format:%A, %d %B, %Y %H:%M:%S']' returned non-zero exit status 128. " Is this normal? Also, for non-specialists, a comment on git may be useful in explaining the code installation.

A l’éditeur seulement:

Recommendation

Publish (easily meets expectations and criteria for this Journal; among top 50%)

---

## Editorial Decision

awaiting_resubmission